# Effects of facial skin pigmentation on social judgments in a Mexican population

**Jaaziel Martínez-Ramírez[1], David Puts[2], Javier Nieto[3], Isaac G-Santoyo[1,4] \***

**1** Faculty of Psychology, Neuroecology Laboratory, National Autonomous University of Mexico, Mexico City, Mexico, **2** Department of Anthropology and Center for Human Evolution and Diversity, The Pennsylvania State University, University Park, College Township, Pennsylvania, United States of America, **3** Faculty of Psychology, Laboratory of Learning and Adaptation, National Autonomous University of Mexico, Mexico City, Mexico, **4** Unidad de Investigación en Psicobiología y Neurociencias, National Autonomous University of Mexico, Mexico City, Mexico

\* isantoyo@psicologia.unam.mx

## Abstract

People quickly and involuntarily form impressions of others based on their facial physical attributes, which can modulate critical social interactions. Skin pigmentation is one of the most variable and conspicuous facial traits among human populations. Empirical evidence suggests that these variations reflect ancestral ecological selective pressures balancing cutaneous vitamin D synthesis with the protection of the dermis from ultraviolet radiation. Nevertheless, skin pigmentation may currently be subject to additional selective pressures. For instance, the colonial era in Central and South America developed a highly stratified society based on ethnic origins, and light skin pigmentation became associated with higher social status and deference. This association could have originated through historical social learning that promoted favorable social perceptions towards individuals with lighter skin color and unfavorable perceptions towards individuals with darker skin color, which could still be present in the perception of current populations. Facial skin pigmentation is also sexually dimorphic, with males tending to exhibit darker skin than females, a difference that could be driven by sexual selection. To explore whether social learning and sexual selection represent additional selective pressures on skin pigmentation, we tested how this facial trait influences fundamental social perceptions in a Mexican population (N = 700, 489 female). We sampled facial images of eight European American males with natural lighter facial skin and eight males from an indigenous pre-Columbian community from Mexico, the Me'Phaa, with natural darker facial skin. We produced stimuli from these images by varying the skin pigmentation while preserving the facial shape. Stimuli were rated on attractiveness, trustworthiness, perceived health, dominance, aggressiveness, and femininity/masculinity. We found that the natural light-skinned faces were perceived as more attractive, trustworthy, and healthy but less dominant than the natural dark faces. Furthermore, by varying the facial skin color in these original groups, we altered the perceptions of them, mainly their attractiveness. These results partially support the hypothesis that dark facial skin color may help males compete for mates. Also, the results strongly support the view that lighter facial skin color became associated with social benefits through social learning in this Mexican population. Our findings, when viewed through the lens of cultural evolution, align with previous

**Data Availability Statement:** All data, stimuli and codes for statistical analyses are openly available through the Open Science Framework: https://osf.io/6gn5t/?view_only=89b534d1095b4abf87411b64fd6d8e08

**Funding:** The present work was part of the degree requirements for JMR Doctorate thesis at the Programa de Maestría y Doctorado en Psicología, UNAM. JMR received a Doctorate Nacional grant from the Consejo Nacional de Ciencia y Tecnología (CONACyT, CVU: 921005), and the work was supported by founding from UNAM-PAPIIT [grant numbers IA209416, IA207019] and CONACYT Ciencia Básica [grant number 241744]. The funders had no role in study design, data collection and analysis, decision to publish, or preparation of the manuscript.

**Competing interests:** The authors have declared that no competing interests exist.

research in social psychology and anthropology. They hold the potential to offer a comprehensive understanding of the origin of this social phenomenon of cultural transmission, which currently plays a role in the formation of racial attitudes, stereotyping, and racial inequality in Mexican and other Latin American populations.

## Introduction

Human faces provide rich information about salient attributes, such as mate quality [1,2], social dominance [3], and fecundity [4]. Hence, people quickly and involuntarily form impressions of others based on their facial appearance [5,6], which in turn may influence important social outcomes [7,8]. Skin pigmentation is one of the most conspicuous and variable facial traits of human populations [9]. The main pigments—melanin, hemoglobin, and carotenoids —absorb and reflect different wavelengths of light within the visible spectrum, and collagen also contributes by scattering light [10,11]. The skin's pigments vary not only between individuals and populations but also between men and women, perhaps because of sexual selection [12]. Natural selection initially favored darker skin as a means to protect against the harmful effects of UV radiation in the tropics. The pressure of selection then shifted toward lighter skin among those humans who spread into higher latitudes with lower levels of UVR [13–15]. Melanin, the principal skin pigment that provides this gradient of light to dark color, acts as a photoprotective filter that reduces the penetration of all wavelengths of light into subepidermal tissue. Under the high UVA levels near the equator, dark skin pigmentation is particularly advantageous for protecting folate and reducing the risk of fatal birth defects. Human migration beyond the tropics reduced the selective advantage of this photoprotective property and selected instead for lower melanization to facilitate cutaneous vitamin D biosynthesis, which is critical to the regulation of many biological processes [16]. Although these events are widely accepted as the main evolutionary events generating human skin pigment variation, this product of natural selection could be currently subjected to other ecological pressures in addition to those that caused its origin.

Cultural transmission is the main means for passing on adaptative behaviors that improve one´s chances for survival and reproduction [17]. It is especially adaptive if a transmitted behavior saves learners the costs of individual learning through trial and error. For instance, cultural transmission can influence mate preferences through two different mechanisms [18]; by reinforcing a preference for a trait that may signal biological quality if that preference significantly contributes to reproductive success, such as offspring quality, or by promoting new preferences for traits that do not appear to have direct links to the bearer's biological quality [19]. In this case, new social preferences may be selected by cultural transmission if the novel trait positively influences social benefits in a particular population [20].

The colonial period in Mexico and all countries of Central and South America led to a highly stratified society based on ethnic origins. During the 300 years of that period, light skin pigmentation, usually present in Spanish colonizers, was one of the most conspicuous phenotypic traits associated with higher social status and hence access to social benefits, such as education and wealth [21]. This racial/color hierarchy system, known as *castes*, placed the European colonizers at the top of the social hierarchy and the indigenous populations at the bottom. It developed deep roots in Latin American society, and even though the *caste* system was officially abolished, there is robust evidence of social stratification based on skin color in

contemporary Mexican society [22], which can be accompanied by racial attitudes, stereotyping and racial inequality [22].

Recent studies have shown that people with darker skin have, on average, lower educational levels, occupational status, and hourly earnings, are more likely to live in poverty, and have fewer chances of social mobility than do people with lighter skin; these differences are significant even after controlling for sociodemographic factors such as age, gender, and parental education [23–25]. In addition, a cultural preference for light skin and other European phenotypical features is also influenced by social media in Western societies, which might reinforce the association between skin color and social status. Hence, if light skin is culturally preferred because it is associated with social benefits, prestige, or deference, a new skin pigmentation signal may have emerged through an extension of historical, sociocultural learning.

Another non-mutually exclusive hypothesis is that facial skin pigmentation could be driven by perceptions shaped by sexual selection. Human skin pigmentation is a sexually dimorphic trait in the sense that men usually have darker skin pigmentation than women [26–28]. The sex difference is due to exposure of skin tissues to differing ratios of androgens to estrogens, particularly at puberty. Testosterone has a stronger effect than estrogen on melanin synthesis and vascularization of the upper dermis [29–31]. In addition, males develop a more intense facultative pigmentation after sun exposure and retain it for longer periods than females whereas female skin lightens faster after a reduction in sun exposure [32]. This sexual dimorphism might influence sexual preferences, since several studies have shown that darker skin in men is perceived as more attractive [33–36]. Therefore, this female preference for darker skin color could reflect ancestral selection pressures shaping perceptions of an androgen-dependent trait. In this case, the basis of this preference may be the honest signaling of mate quality in terms of male immune function (i.e., immunocompetence handicap hypothesis), vigor, or other physiological costs of developing this trait [37–39].

Although the effects of skin pigmentation on social perceptions is an active area of research in social psychology, prior research has focused mainly on racial attitudes and generally does not employ an evolutionary framework. For instance, Hagiwara et al. [40] evaluated affective reactions and other social attributions towards African American men by manipulating facial shape features (nose and lips) while controlling for skin tone. Similar studies have examined the role of skin tone and facial shape features in racial attitudes through similar methods [41,42]. Moreover, most experimental studies of facial perception have been carried out on participants from WEIRD (Western, educated, industrialized, rich, and democratic) societies, thus limiting the generalizability of findings [43]. Finally, prior studies in non-WEIRD Latin American samples have implemented a social theoretical framework with a non-experimental design to explore correlations between skin tone and social attributes, such as status or labor hierarchy [44].

In this study, we used an experimental design with a Mexican population to determine whether facial skin color can influence certain social perceptions: perceived attractiveness, perceived trustworthiness, perceived health, perceived dominance, perceived aggressiveness, and perceived femininity/masculinity [45–47]. To that end, we performed a crossover experiment on men's faces, selecting two groups of faces with contrasting innate skin pigmentation: the first composed of faces with natural light facial skin tone, selected from a European American population, and the second composed of faces with a darker facial skin tone, selected from a Mexican pre-Columbian population, self-identified as Me'Phaa. Using these two groups, we created six sets of facial stimuli by varying the natural skin pigmentation while preserving the facial shape. Finally, we presented all stimuli to be evaluated in the six social dimensions previously mentioned.

We hypothesize that if social learning is acting on the perceptions of facial skin pigmentation, we expect that the lighter-skinned group would be perceived as more attractive, more trustworthy, and healthier because positive evaluations of these attributes strongly promote pro-sociality and social benefits, but less dominant and less aggressive because dominance and aggressiveness, in contrast, are considered to be potentially harmful and may lead to avoidance of social interactions [48]. However, if skin pigmentation more strongly reflects ancestral selection-shaping perceptions of an androgen-dependent trait, then we would expect that faces with darker skin would be perceived not only as more attractive and healthier but also as more dominant, aggressive, and masculine due to the association between these last three attributes and the behavioral and physiological effects of androgens [49,50]. In both cases, if differences in skin pigmentation are driving these effects, we expect that the experimental manipulation of facial skin color would change the previous patterns in the perceptions of both the European American and Me´Phaa faces. Finally, if neither light nor dark skin color affects any social perception of the original faces and the experimentally exchanged stimuli, then skin color would play no role in either social selection or the sexual selection of dimorphic traits.

## Material and methods

### Participant raters

A total of 489 Mexican adult women (mean age = 25.4 years, SD = 8.6 years) and 211 Mexican adult men (mean age = 26.5 years, SD = 9.1 years) participated as raters in this study. Self-reported ethnicities were 84.0% Mestizo, 6.3% European, 3.9% Indigenous (particular groups not specified), 0.3% African, and 6.3% Other. A total of 485 participants reported living in Mexico City, 95 in the surrounded metropolitan area, and the rest in different states of Mexico. All the raters were recruited online through a board advertisement from the Department of Psychology at the National Autonomous University of Mexico.

### Skin color facial stimuli

We first created two contrasting light and dark facial skin prototypes by selecting 30 facial photographs of young European American men and 30 facial photographs of young Me'Phaa Mexican Indigenous men from larger photograph sets. (See below the criteria of selection for both groups of photographs). The European American faces were selected from a set of facial photographs of the Department of Anthropology at Pennsylvania State University [51]. Facial photographs were taken from a distance of 2m using a 12-megapixel Olympus E-300 camera with a mounted flash at a resolution of 1200 × 1000 pixels. The facial photographs of the Me'Phaa were selected from a collection of facial photographs from the NeuroEcology Lab at the National Autonomous University of Mexico. These facial photographs were taken from a distance of 2m using a 21-megapixel Nikon 5500 camera illuminated by two studio strobes (Lusana Studio, 400w) with reflective umbrellas. The strobes were placed 1.85 m from the target on each side and lights were arranged at 40˚ angles. In both sets of facial photographs, participants were asked to remove any earrings, glasses, or facial jewelry, and to maintain a neutral expression.

The two groups of facial photographs (European American and Me´Phaa) were selected in order to obtain two samples of faces whose pigmentation was contrasting enough to avoid an overlapping effect: one group with lighter skin and color ranges typical for the European American average, and another group with darker skin and color ranges typical for the Indigenous Mexican average, and because both ranges of skin color are within the typical range of colors in the Mexican population. Therefore, in order to evaluate if the facial photographs in both groups met these criteria, we measured the skin color values in CIELab color space.

Following procedures reported in previous studies [52,53], we first standardized the facial images by color calibration, then we took color samples of three separate patches located on the forehead and on each cheek and calculated the average CIELab values using Photoshop CS6. Lightness ($L^*$) value was the parameter that changed the most between the ethnic groups we sampled [54].Therefore, we focused on this parameter as a representative value of skin color differences between the two selected groups. Here, European American ($L^*$ mean = 61.08, SD = 6.42) and Me'Phaa ($L^*$ mean = 49.84, SD = 4.05) mean lightness were statistically different ($t$ (58) = -8.07, $p < 0.01$), and lightness values for both groups were in the ranges of White/European and Hispanic ethnicities reported in previous studies [55,56].

From the 60 facial photographs selected, we created two composite faces that averaged the skin color of the 30 original faces from each group using the software Psychomorph [57]. The averaged composite face of the European American men was the light skin prototype, while the composite face of the Me'Phaa men was the dark skin prototype. Next, we randomly selected eight faces from the European American male set and eight from the Me'Phaa male set and created two different versions of each face, one displaying light skin and one displaying dark skin. These two versions were created by adding or subtracting 70% of the skin color differences between the light and dark facial skin prototypes previously created while leaving facial shape unchanged. Finally, we repeated the skin color measurement process (see above) in both groups of the new composite faces (light skin color transformed and dark skin color transformed faces). The average lightness value was $L^* = 66.74$, SD = 1.37 in both light skin color transformed versions (i.e., Natural Me´phaa and Natural European American faces) and $L^* = 52.75$, SD = 2.58 in both dark skin color transformed versions. The method we used for skin color manipulation slightly increased the lightness values with respect to the values of the natural faces, the increases were not statistically significant for any group ($t$ (44) = -0.52, $p = 0.6; t$ (44) = -0.43, $p = 0.6$), and both groups stayed within the ranges reported for White/European and Hispanic ethnicities in previous studies [55,56]. At the end of this process, we obtained three versions for each of the eight European American and eight Me'Phaa faces: a natural unmodified version (NAT), a light skin color transformed version (LSCt), and a dark skin color transformed version (DSCt; Fig 1), for a total of 48 facial stimuli.

## Facial perception rates

To obtain the ratings, we divided the 48 stimuli into eight sets of six stimuli each. The stimuli were assigned to each set counterbalancing by group (European American and Me'Phaa) and version (NAT, LSCt, and DSCt), and we made sure that two or more versions of the same face were not included together in the same set. Given that the experimental transformation tended to homogenize facial skin color and texture, two features that affect perceptions of human faces [58,59], we decided to include the natural version of each face (NAT) to be able to compare the ratings of the natural versions with the ratings of the concordant skin color transformed version; that is, we compared the Natural versions with the Light skin color transformed versions in the European American faces and the Natural versions with the dark skin color transformed versions in the Me'Phaa faces. By doing this we could measure the effect of the color and texture homogenization.

The facial evaluations were performed online at the webpage labneuroecologiacog.com. Each participant was randomly assigned to one of the eight facial stimuli sets and was asked to rate each face separately on a 9-point Likert scale according to the following perceptions: attractiveness (1 = not attractive at all, 9 = very attractive); trustworthiness (1 = not trustworthy at all, 9 = very trustworthy); perceived health (1 = not healthy at all, 9 = very healthy); dominance (1 = not dominant at all, 9 = very dominant); aggressiveness (1 = not aggressive at all,

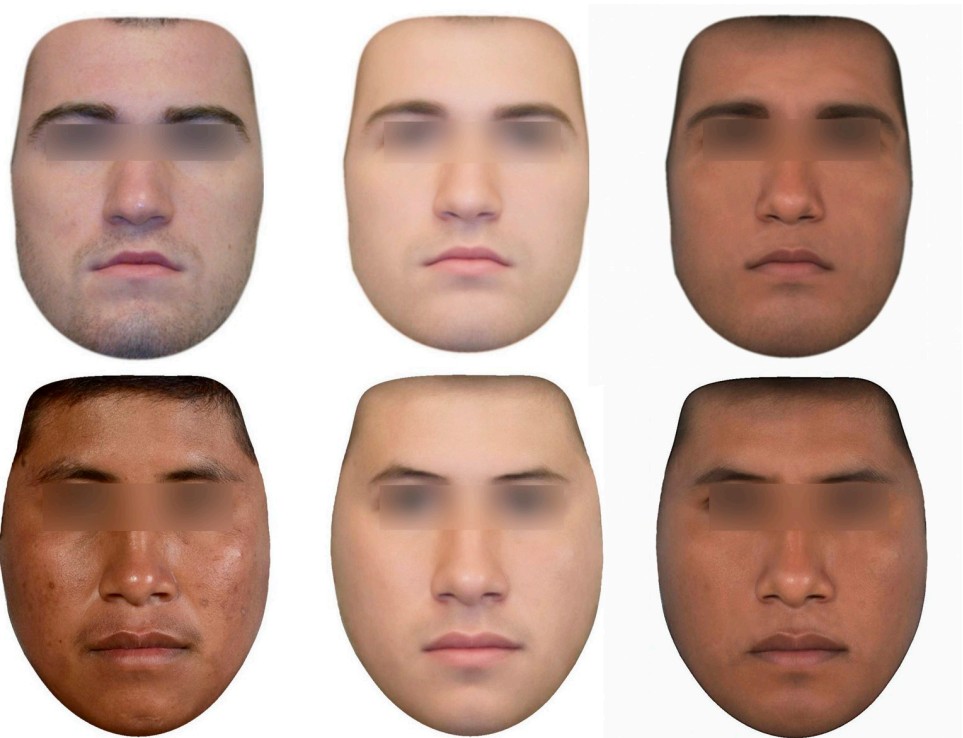

**Fig 1. Example stimuli.** Three versions of the same facial stimuli of a European American male (up) and a Me'Phaa male (bottom). Natural unmodified versions (left), Light Skin Color transformed versions (middle), and Dark Skin Color transformed versions (right).

9 = very aggressive); and femininity/masculinity (1 = very feminine/not masculine at all, 9 = very masculine/ not feminine at all). For all trials, we randomized both the order of facial stimuli and the order of the questions about the perceived qualities. In addition, to standardize the response for attractiveness in men and women, we asked the men participants, *"How attractive does this face look for a woman?"* (For similar methods see [60]). Finally, raters completed a socio-demographic survey and were asked about their sexual orientation, on a 5-point Likert scale, from entirely same-gender to entirely opposite-gender orientation. As a result, 610 raters (437 women and 173 men) identified themselves as opposite-gender oriented/heterosexual, 42 (15 women and 27 men) identified as same-gender sexual orientation, 35 (28 women and 7 men) identified as bisexual, and 13 (9 women and 4 men) reported not having any sexual orientation. We controlled the sexual orientation information in the statistical analysis (see below). In addition, self-perceived skin color may have been influencing the rater's responses [23,61]. Hence, raters were asked to rate their own skin color using the PERLA color palette. The PERLA color palette is a scale of 11 skin tones (1 = lightest, 11 = darkest) that was made to provide an easy-to-use portrayal of the typical skin tones found in regions of Latin America [62]. Similar to previous studies on the Mexican population [24], we found that 77. 6% of the raters´ responses were concentrated in the middle levels of the scale (4–6), while the rest were at the extremes: 12.7% in the lightest skin tones (1–3) and 9.7% in the darkest ones (7–10; see S1 Fig). Finally, all raters signed a digital informed consent form before evaluating facial stimuli, and all procedures were approved by the appropriate National Autonomous University of Mexico ethics committee (FPSI/CE/01/2016) and are in compliance with

Mexican legislation and the Code of Ethical Principles for Medical Research Involving Human Subjects of the World Medical Association (Declaration of Helsinki).

## Statistical analysis

To determine whether the experimental manipulation of facial skin color modified the perceived qualities, we fitted Poisson generalized linear mixed models (GLMM) with the log link function. We decided to fit Poisson GLMMs because, unlike other studies where the ratings for each facial stimulus are averaged, here all dependent variables come from discrete raw ratings. Moreover, during the data exploration we observed a positively skewed counts-type distribution with most of the observations at the lower values in some of the facial perceptions (i.e., attractiveness perception). We ran one global model per each perception evaluated. Global models included as the dependent variable any of the following facial perceptions: attractiveness, trustworthiness, perceived health, dominance, aggressiveness, or femininity/masculinity. Fixed predictors for all global models included the ethnic group of each evaluated face (i.e., European American or Me'phaa faces), the natural and transformed skin color versions (i.e., NAT, LSCt or DSCt), sex of raters, and the interaction between ethnic group and skin color versions. To account for the potential non-independence of data, we included a random intercept for rater ID, a random intercept for facial photograph ID (the individual code of each original face used), and a random slope for skin color transformation on facial photograph ID.

To evaluate the effect of rater´s sexual preferences, we ran models for responses from all participants and models for responses from heterosexual participants only. Next, each global model was then reduced to obtain a minimum adequate model using the Akaike Information Criterion (AIC). We selected the models with the lowest AIC values and a ΔAIC higher than two units from the second most supported model [63]. When the difference between the best-supported model and the subsequent model was Δ<2, we followed the statistical principle of parsimony [64], and hence we selected the minimum adequate model as the one with the lowest AIC and the fewest parameters (Table 1). Finally, once we obtained minimum adequate models, model diagnostics were performed, and we checked residual distribution and linearity of residuals in each model. In order to check for any effect of other factors not included in the models [65,66], we plotted Pearson residuals of each selected model against the rater´s self-reported ethnicities and the PERLA scale responses for self-perceived skin color. Since the plots did not show any clear effect, we did not include either of the two factors as fixed predictors. All analyses were performed using the *lme4* [67] and *bbmle* [68] packages in R, version 4.0.2 [69].

## Results

### Perceived attractiveness

Perceived attractiveness differed significantly between the two groups when they had their natural skin color ($z = 6.51$, $p < 0.01$). The natural lighter-faced group (i.e., composed of natural European American faces) was rated 1.70 times more attractive ($M = 4.30$, 95% CI [3.83–4.84]) than the natural darker-faced group (i.e., composed of natural Me´phaa faces; $M = 2.50$, 95% CI [2.21–2.82], Table 2). Furthermore, there were effects when natural skin color was changed within each of the two groups. Lightening the natural darker group (i.e., Me'Phaa faces) made them more attractive ($M = 3.75$, 95% CI [3.42–4.11], $z = 7.42$, $p < 0.01$), and darkening the opposite group (i.e., European American faces) made them less attractive ($M = 3.30$, 95% CI [3.02–3.60], $z = -6.25$, $p < 0.01$). In addition, we observed an effect of skin color and texture homogenization due to experimental manipulation. Both groups became slightly more

**Table 1. Information criteria for the best fit models.**

|  | df | AICc | ΔAICc | Wi(AICc) |
|---|---|---|---|---|
| *(a) Perceived attractiveness* | | | | |
| **Group + Color + Rater´s sex + Group:Color** | **14** | **14331** | **0.000** | **0.999** |
| Group + Color + Rater´s sex | 12 | 14346 | 15.053 | 0.001 |
| Group + Color + Group:Color | 13 | 14348 | 16.957 | 0.000 |
| *(b) Perceived trustworthiness* | | | | |
| **Group + Color** | **11** | **15167** | **0.000** | **0.506** |
| Group + Color + Rater´s sex | 12 | 15169 | 1.778 | 0.208 |
| Group + Color + Group: Color | 13 | 15170 | 2.596 | 0.138 |
| *(c) Perceived health* | | | | |
| **Group + Color + Group:Color** | **13** | **15561** | **0.000** | **0.707** |
| Group + Color + Rater´s sex + Group:Color | 14 | 15563 | 1.892 | 0.274 |
| Group + Color | 11 | 15570 | 9.186 | 0.007 |
| *(d) Perceived dominance* | | | | |
| **Group + Rater´s sex** | **10** | **15485** | **0.000** | **0.493** |
| Group + Color | 11 | 15488 | 2.685 | 0.129 |
| Group + Color + Group: Color | 13 | 15489 | 3.137 | 0.103 |
| *(e) Perceived aggressiveness* | | | | |
| **Group + Color + Group:Color** | **13** | **16625** | **0.000** | **0.663** |
| Group + Color + Rater´s sex + Group:Color | 14 | 16626 | 1.768 | 0.274 |
| Group + Color | 11 | 16630 | 5.479 | 0.043 |
| *(f) Perceived masculinity* | | | | |
| **Group + Color + Group:Color** | 14 | 15486 | 0.000 | 0.482 |
| Group + Color + Rater´s sex + Group:Color | **13** | **15486** | **0.035** | **0.474** |
| Group + Color | 11 | 15492 | 6.367 | 0.020 |

Information criteria and fixed factors included for the three best fit models for: (a) Perceived attractiveness, (b) Perceived trustworthiness, (c) Perceived health, (d) Perceived dominance, (e) Perceived aggressiveness, and (f) Perceived masculinity.

Models are in descending order. ΔAIC is the change in AIC between each model and the best model. Akaike weights are conditional probabilities for each model being the best model. Models in bold are the selected ones.

attractive when skin color and skin texture were homogenized (Me'Phaa faces: $M = 2.94$, 95% CI [2.68–3.21]; European American faces: $M = 4.77$, 95% CI [4.36–5.22]). However, this effect was significant for darker-faced group (i.e., Me'Phaa faces; $z = -3.23$, $p < 0.01$) but not for lighter-faced group (i.e., the European American faces; $z = -2.00$, $p = 0.34$; Fig 2). Finally, we observed a significant main effect of rater´s sex ($z = 4.36$, $p < 0.01$); all the faces seemed more attractive to the male raters ($M = 3.76$, 95% CI [3.48–4.06]) than to the female ones ($M = 3.27$, CI 95% [3.06–3.49]). The direction of the effects and statistical significance were not affected when including the rater´s sexual preference (S1 Table).

## Perceived trustworthiness

The natural lighter-faced group was rated 1.13 times more trustworthy ($M = 4.75$, 95% CI [4.55–4.96], $z = 5.19$, $p < 0.01$) than the natural darker-faced group ($M = 4.21$, 95% CI [4.03–4.40], Table 2). In addition, lightening the natural darker-faced group (i.e., Me'Phaa faces) increased their perceived trustworthiness ($M = 4.48$, 95% CI [4.30–4.67], $z = 3.26$, $p < 0.01$) whereas darkening the natural lighter-faced group (i.e., European American faces) had no effect ($M = 4.89$, 95% CI [4.70–5.08], $z = 1.44$, $p = 0.15$). Skin color and texture

**Table 2. Perception estimates with 95% confidence intervals.**

| | Me'Phaa | | | European American | | |
|---|---|---|---|---|---|---|
| | **NAT** | **LSCt** | **DSCt** | **NAT** | **LSCt** | **DSCt** |
| | *Est. [95%CI]* | *Est. [95%CI]* | *Est. [95%CI]* | *Est. [95%CI]* | *Est. [95%CI]* | *Est. [95%CI]* |
| **a) Perceived attractiveness** | 2.50 [2.21–2.82] | 3.75 [3.42–4.11] | 2.94 [2.68–3.21] | 4.30 [3.83–4.84] | 4.77 [4.36 5.22] | 3.30 [3.02–3.60] |
| **b) Perceived trustworthiness** | 4.21 [4.03–4.40] | 4.48 [4.30–4.67] | 4.33 [4.16–4.51] | 4.75 [4.55–4.96] | 5.06 [4.86–5.27] | 4.89 [4.70–5.08] |
| **c) Perceived health** | 4.28 [3.83–4.79] | 5.51 [5.19–5.84] | 5.21 [4.85–5.60] | 5.10 [4.56–5.69] | 5.96 [5.62–6.31] | 5.05 [4.70–5.43] |
| **d) Perceived dominance** | 5.23 [4.85–5.64] | �follower | �follower | 4.31 [3.99–4.66] | �follower | �follower |
| **e) Perceived aggressiveness** | 5.43 [5.17–5.69] | 5.05 [4.85–5.26] | 5.39 [5.17–5.62] | 5.10 [4.85–5.35] | 4.79 [4.59–4.99] | 4.56 [4.37–4.77] |
| **f)z Perceived masculinity** | 6.18 [5.79–6.60] | 5.15 [4.83–5.49] | 5.97 [5.59–6.37] | 5.14 [4.80–5.50] | 4.83 [4.53–5.15] | 4.93 [4.61–5.27] |

Estimates (Est.) with 95% confidence intervals (CI) of the fixed predictors of the models for: (a) Perceived attractiveness, (b) Perceived trustworthiness, (c) Perceived health, (d) Perceived dominance, (e) Perceived aggressiveness, and (f) Perceived masculinity.

*Note*: (�follower) The Perceived dominance best fit model did not include the skin color transformation predictor, therefore, the estimates shown represent the main effect of the natural skin color group only.

homogenization did not increase scores for the natural darker faces (DSCt Me'Phaa faces $M = 4.89$, 95% CI [4.70–5.08], $z = -1.43$, $p = 0.70$), but did increase scores for the natural lighter faces (LSCt European American faces $M = 5.06$, 95% CI [4.86–5.27], $z = -3.26$, $p = 0.01$; Fig 3). Finally, there was no effect of rater´s sex, as this predictor was not present in the minimum adequate model selected by AIC (Table 1). The direction of the effects and statistical significance were not affected when the rater´s sexual preference was included (S1 Table).

## Perceived health

As with perceived attractiveness and trustworthiness, we found a significant difference in the perceived health scores between the natural darker and lighter faces ($z = 2.20$, $p = 0.02$). On average, natural lighter faces were perceived as being 1.19 times healthier ($M = 5.10$, 95% CI [4.56–5.69]) than natural darker faces ($M = 4.28$, 95% CI [3.83–4.79], Table 2). When facial skin color was changed within each ethnic group, the changes had different effects on perceptions of the two groups. Lightening the natural darker faces increased their perceived health ($M = 5.51$, 95% CI [5.19–5.84], $z = 4.96$, $p < 0.01$), whereas darkening the lighter faces had no effect ($M = 5.05$, 95% CI [4.70–5.43], $z = 0.20$, $p = 0.99$). Perceived health increased slightly for both groups when skin color and skin texture were homogenized (European Americans: $M = 5.96$ (95% CI [5.62–6.31], $z = -3.12$, $p = 0.02$; Me'phaa people: $M = 5.21$, 95% CI [4.85–5.60], $z = 4.38$, $p < 0.01$; Fig 4). Finally, there was no effect of the rater´s sex, as this predictor was not present in the minimum adequate model selected by AIC (Table 1). The direction of the effects and statistical significance were not affected when the rater´s sexual preference was included (S1 Table).

## Perceived dominance

Unlike the previous qualities, perceived dominance was lower for natural lighter faces ($M = 4.31$, 95% CI [3.99–4.66], $z = -3.63$, $p < 0.01$) than for natural darker faces ($M = 5.23$, 95% CI [4.85–5.64], Table 2). Moreover, in contrast to perceived attractiveness, all faces received a

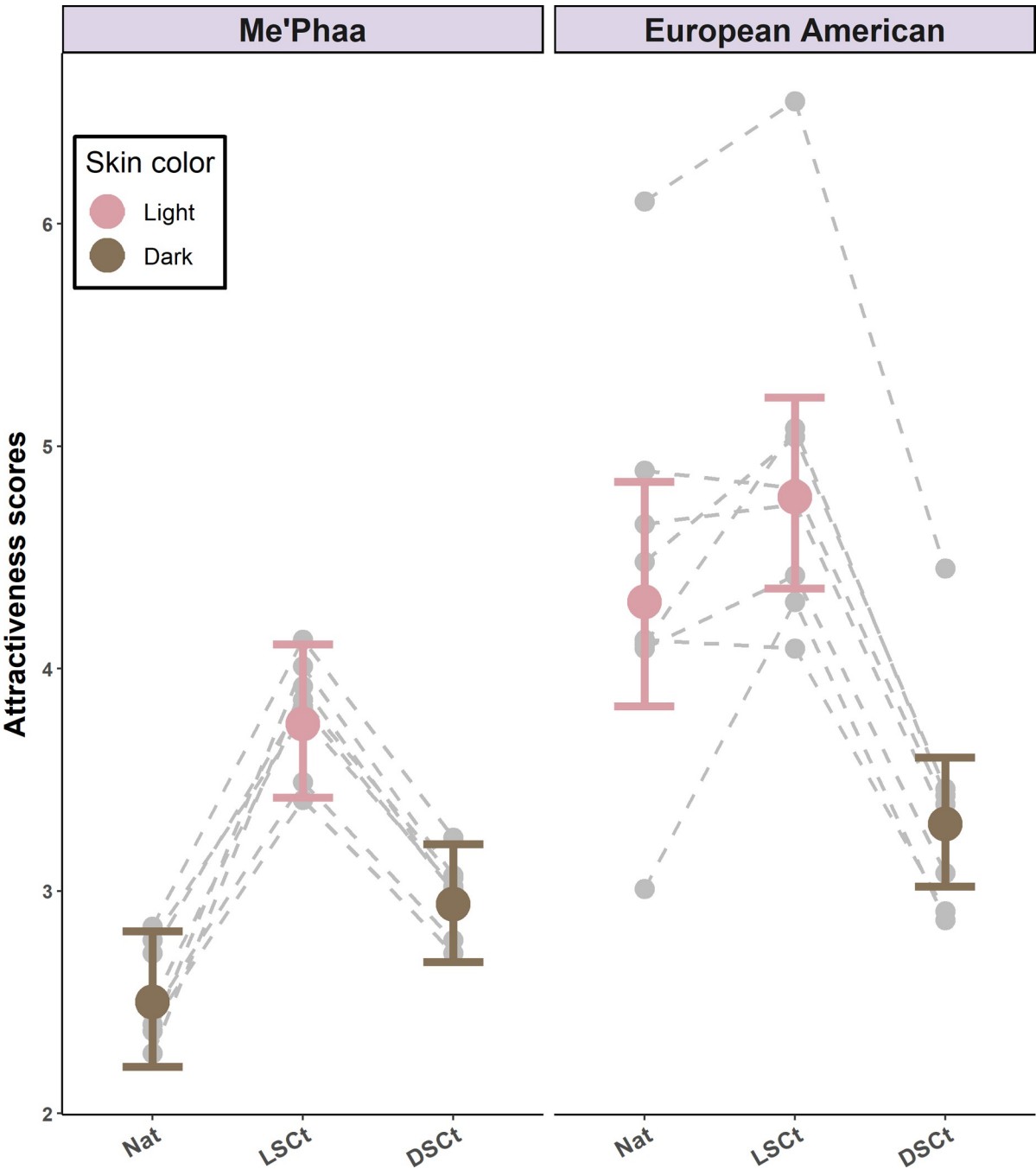

**Fig 2. Perceived attractiveness estimates for the Me'Phaa and European American faces within each color version.** Estimates for the Natural versions (NAT), Light skin color transformed versions (LSCt), and Dark skin color transformed versions (DSCt) of the Me'Phaa (left) and European American (Right) faces. Error bars show 95% Confidence Intervals. Grey points and dashed lines indicate the random effect estimates for each Face ID within each color version.

lower dominance rating from male raters ($M = 4.63$, 95% CI [4.34–4.93], $z = -2.24$, $p = 0.02$) than from female raters ($M = 4.88$, 95% CI [4.61–5.16]). Finally, we were unable to determine any effect from lightening, darkening, or homogenizing the faces, since these predictors were

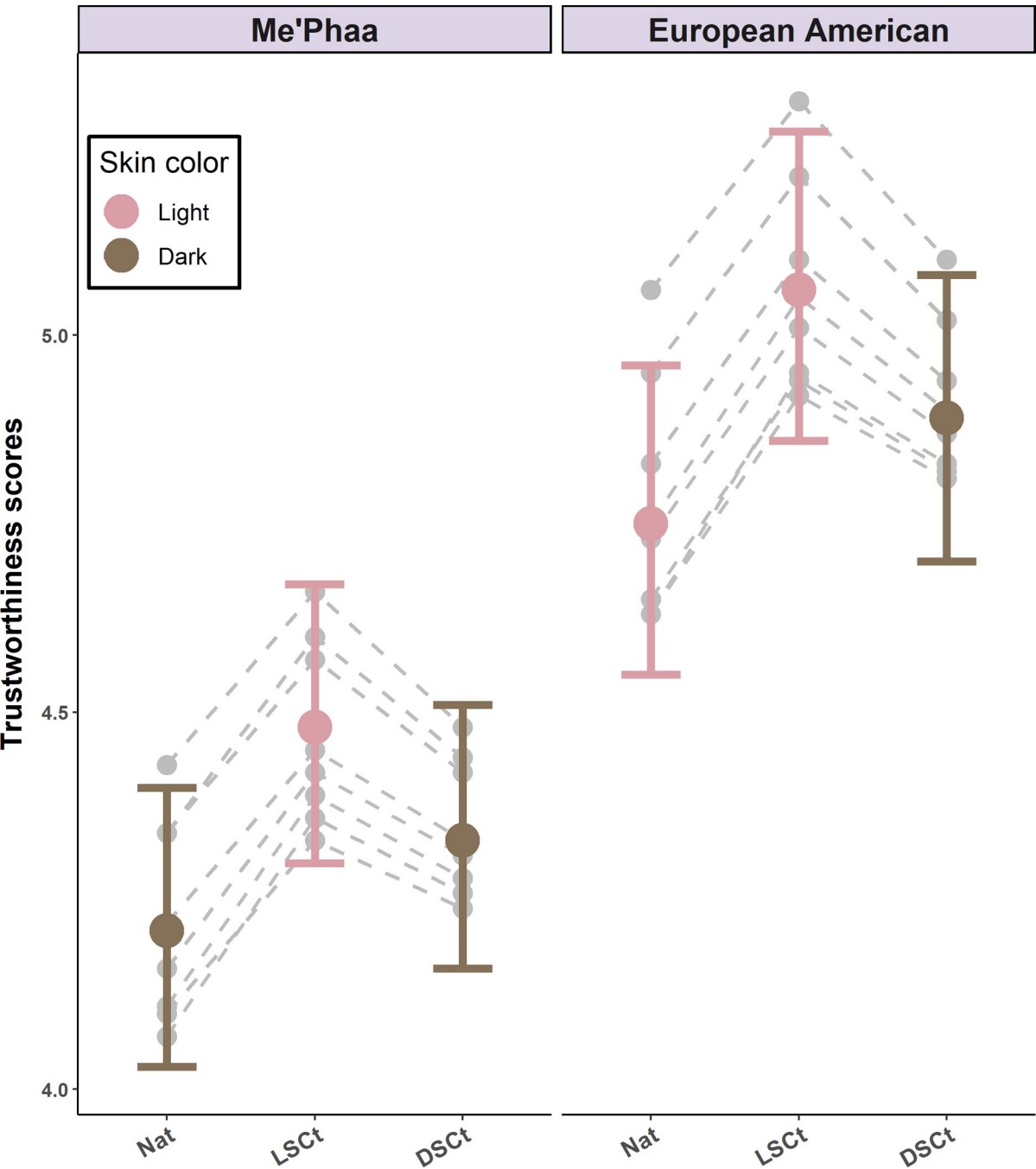

**Fig 3. Perceived trustworthiness estimates for Me'Phaa and European American faces within each color version.** Estimates for the Natural versions (NAT), Light skin color transformed versions (LSCt), and Dark skin color transformed versions (DSCt) of the Me'Phaa (left) and European American (Right) faces. Error bars show 95% Confidence Intervals. Grey points and dashed lines indicate the random effect estimates for each Face ID within each color version.

not present in the minimum adequate model selected by AIC (Table 1). Statistical significance of these results did not change in the model that included all raters (S1 Table).

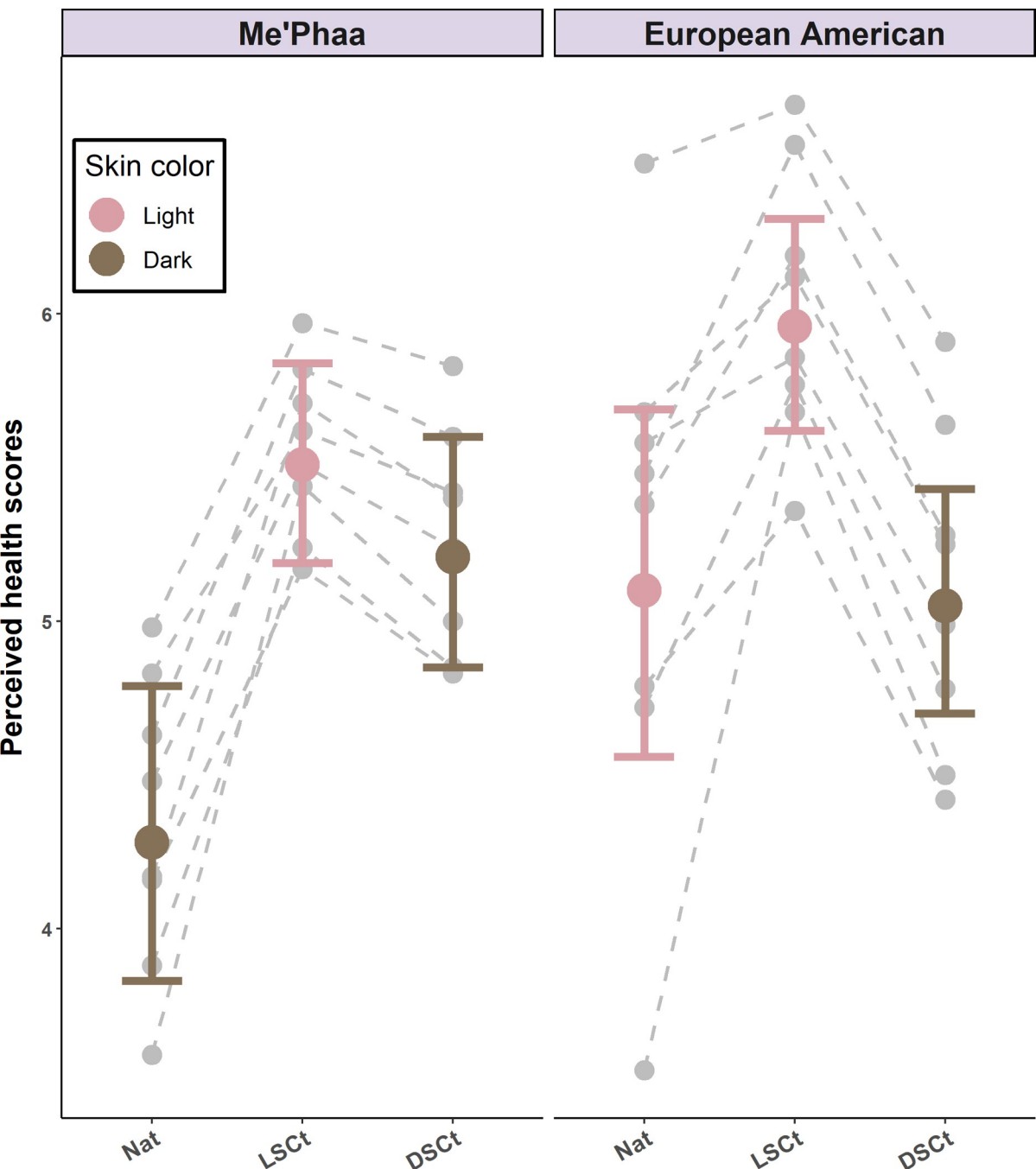

**Fig 4. Perceived health estimates for Me'Phaa and European American faces within each color version.** Estimates for the Natural versions (NAT), Light skin color transformed versions (LSCt), and Dark skin color transformed versions (DSCt) of the Me'Phaa (left) and European American (Right) faces. Error bars show 95% Confidence Intervals. Grey points and dashed lines indicate the random effect estimates for each Face ID within each color version.

### Perceived aggressiveness

Although perceived aggressiveness was lower for the lighter faces, the difference was not significant ($z$ = -1.83, $p$ = 0.06). Nevertheless, lightening the darker faces made them seem less aggressive ($M$ = 5.05, 95% CI [4.85–5.26], $z$ = -2.76, $p$ < 0.01) but darkening the lighter faces

also made them seem less aggressive ($M = 4.56$, 95% CI [4.37–4.77], $z = 4.17$, $p < 0.01$, Table 2). Finally, perceived aggressiveness did not change for either group when skin color and skin texture were homogenized (Me'Phaa faces $M = 5.39$, 95% CI [5.17–5.62]; $z = -0.23$, $p = 0.82$; European American faces $M = 4.79$, 95% CI [4.59–4.99]; $z = 2.35$, $p = 0.17$, Fig 5.), and there was no effect of the rater´s sex, as this predictor was not present in the minimum adequate model selected by AIC (Table 1). Finally, the direction of the effects and statistical significance were not affected when the rater´s sexual preference was included (S1 Table).

## Perceived masculinity

As with perceived dominance, perceived masculinity was lower for natural lighter faces ($M = 5.14$, 95% CI [4.80–5.50], $z = -3.95$, $p < 0.01$ than for natural darker faces ($M = 6.18$, 95% CI [5.79–6.60], Table 2). Moreover, lightening the darker faces made them seem less masculine ($M = 5.97$, 95% CI [5.59–6.37], $z = -5.59$, $p < 0.01$), but darkening the lighter faces had no effect ($M = 4.93$, 95% CI [4.61–5.27], $z = 1.20$, $p = 0.83$). In addition, perceived masculinity did not change for either group when skin color and skin texture were homogenized (DSCt Me'Phaa faces $M = 5.97$, 95% CI [5.59–6.37]; $z = 1.05$, $p = 0.90$; LSCt European American faces $M = 4.83$, 95% CI [4.53–5.15]; $z = 1.82$, $p = 0.45$, Fig 6). Finally, there was no effect of the rater´s sex, as this predictor was not present in the minimum adequate model selected by AIC (Table 1). The direction of the effects and statistical significance was not affected when the rater´s sexual preference was included (S1 Table).

## Discussion

Our findings suggest that facial skin pigmentation influences social judgments in a Mexican population. Based on our results, we could distinguish two groupings of social perceptions that facial skin pigmentation affects in opposite directions. In comparison with natural Me'Phaa faces, natural European American faces were perceived not only as more attractive, healthier, and more trustworthy but also as less dominant and less masculine. They were furthermore perceived as less aggressive, but not significantly. Moreover, the perceptions grouped in these two dimensions correlate more within than between dimensions (S2 Table.). The apparent grouping of social perceptions found here resembles the valence-dominance model of facial perception proposed by Oosterhof and Todorov [70] and recently discussed by Jones et al. [6]. The valence-dominance model suggests that humans make impressions of others based on their facial appearance in the dimensions of valence and dominance. The valence dimension is related to facial judgments such as trustworthiness and attractiveness; hence, higher rates of this dimension could promote positive social interactions, for example mate opportunities. The dominance dimension, in turn, is related to perceptions such as aggressiveness, dominance, and masculinity, which could inform about the physical or psychological capability to cause harm; hence, higher values of this dimension could negatively affect social interactions in a modern context [8]. Here, we found that natural light-skinned faces were rated more highly on all three aspects of social valence (attractiveness, healthiness, trustworthiness). In contrast, natural dark-skinned faces were rated more highly on two aspects of social dominance (dominance, masculinity).

In the context of our first hypothesis, these results support the idea that sociocultural learning favors the function of facial skin pigmentation as a mediator of social perceptions. Our Mexican participants showed a robust positive association of pro-social judgments (i.e., perceived attractiveness, perceived trustworthiness, and perceived health) with light skin pigmentation. During the Mexican colonial period, light skin pigmentation was one of the most conspicuous phenotypic traits associated with a higher social status and access to different

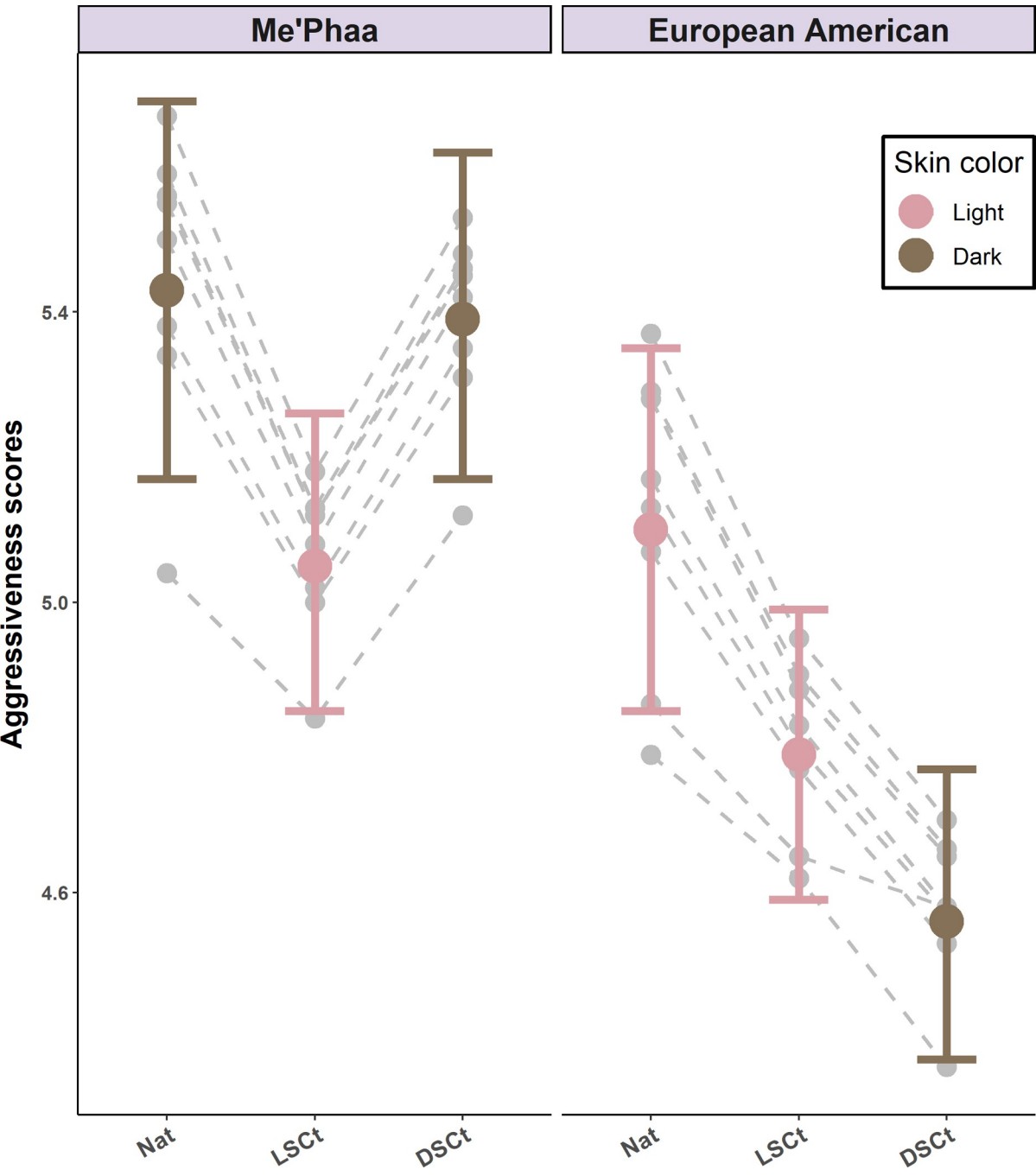

**Fig 5. Perceived aggressiveness estimates for Me'Phaa and European American faces within each color version.** Estimates for the Natural versions (NAT), Light skin color transformed versions (LSCt), and Dark skin color transformed versions (DSCt) of the Me'Phaa (left) and European American (Right) faces. Error bars show 95% Confidence Intervals. Grey points and dashed lines indicate the random effect estimates for each Face ID within each color version.

benefits, such as education and wealth [21]. Learning can reinforce a preference for a trait that may signal social benefits, even when the selected trait is not linked initially to any reproductive or survival benefit [19]. Therefore, a learned association over several generations between a physical trait—such as light skin—and deference might not only establish a socially positive

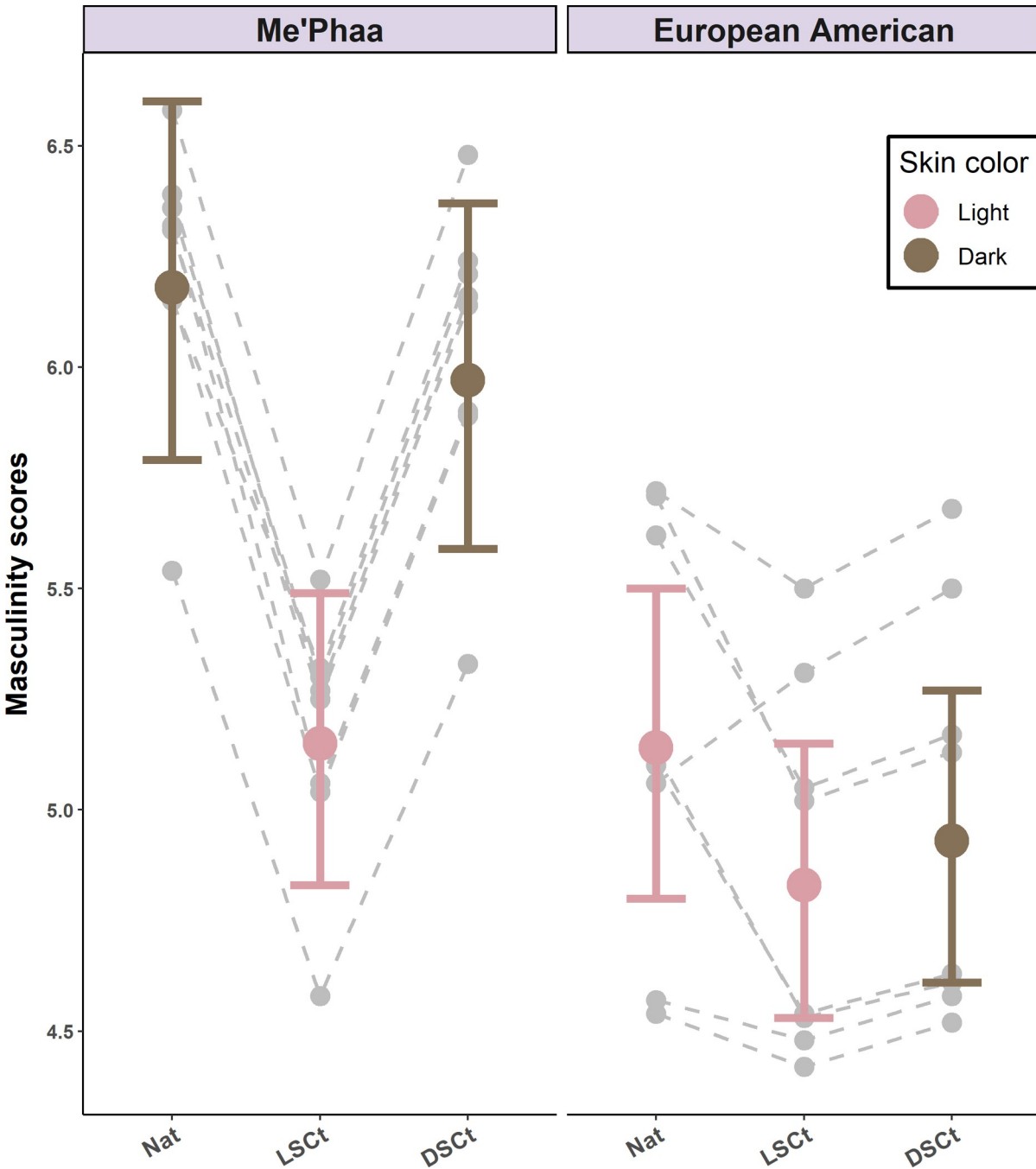

**Fig 6. Perceived masculinity estimates for the Me'Phaa and European American faces within each color version.** Estimates for the Natural versions (NAT), Light skin color transformed versions (LSCt), and Dark skin color transformed versions (DSCt) of the Me'Phaa (left) and European American (Right) faces. Error bars show 95% Confidence Intervals. Grey points and dashed lines indicate the random effect estimates for each Face ID within each color version.

perception of the population that shares this trait, but also give rise to racial stereotypes that negatively affect other coexisting populations, and which reinforce the learning. Although our methodological design did not explicitly provide information about the ethnicity of the stimuli subjects, it is possible that the stereotypes derived from this strong associative learning might

have acted as the driving force influencing the social scores. In Mexico, darker skin color suggests that individuals are more indigenous, and consequently considered more socially unfavorable. Furthermore, several studies have shown cultural preferences for light skin pigmentation and other European phenotypical traits and within-group discrimination based on skin color, a social phenomenon known as colorism [71]. In addition, in Mexico and other Latin American countries, lightness of skin correlates with years of schooling, hourly earnings, occupational status, poverty, and social mobility [23–25]. Therefore, the association of prosocial attributes with light skin pigmentation we observed may be the result of a preference that emerged as an adaptive strategy to deal with social pressures during the colonial period, which nowadays is reinforced by social learning and cultural mechanisms such as colorism, or the influence of Western societies on social media.

We also found that the dark-skinned faces were rated as more dominant and masculine. However, in contrast to our second prediction, they were not rated as more attractive or healthy. These results could indicate that skin pigmentation, as a trait mediated by androgens, could inform the viewer about the physical or psychological capability to cause harm by the bearer of the signal [72,73]; therefore, this signaling function might be relevant in sexual selection through male-male competition instead of female choice[74]. Similar results have been found in previous studies where androgen-dependent sexually dimorphic traits, such as facial shape [75], voice pitch [76,77], or height [78], more strongly affect the perception of dominance, aggressiveness, or masculinity rather than attractiveness.

The human face embodies a variety of signals and cues [79], and even though humans process faces in an integrative manner, this process follows a hierarchy of relevance in which different facial features are more important than others, depending on the perceptions evaluated [80,81]. In this way we found that some social perceptions were more affected by skin color than others. For example, lightening the Me'Phaa faces made them seem more attractive, healthier, and more trustworthy. Conversely, darkening the European American faces made them seem less attractive. Other perceived qualities, however, did not change systematically when facial skin color was changed. Perceived dominance was not affected by manipulations of facial skin color within either group. Lightening the Me'Phaa faces decreased their perceived masculinity and aggressiveness, but darkening the European American faces did not increase their perceived masculinity and aggressiveness. These results are consistent with studies on the role of skin color in certain perceptions, such as attractiveness [59,82], which show that skin color might contribute more than facial shape. Moreover, recent studies have shown that trustworthiness judgments rely on emotional expressions [8] or facial resemblance [83], and that perceptions of dominance, masculinity, and aggressiveness seem to rely more strongly on facial shape [84–88]. Yet another possibility is that skin color interacts with facial shape in some of these perceptions, and it becomes relevant only when facial shape strongly elicits the perception. This may explain why lightening of skin color reduced the perceived masculinity and aggressiveness for the Me´Phaa faces, which were initially rated as more masculine and dominant, but darkening had no effect on the European American faces.

Another possibility is that there were systematic differences in the way that the experimental manipulation of skin color affected the European American and Me´Phaa faces. In the statistical analysis we fitted random slopes for skin color transformation on each facial photograph ID to reduce the standard error caused by these differences; however, it is important to note that any manipulation presents a risk of such artifacts. Moreover, we not only lightened and darkened facial color but also homogenized it by removing such texture-related aspects as spots, freckles, or wrinkles [59,89,90]. Comparing the two color transformed versions of each group, that is the ones with texture and color homogenized, we observed, in general, the main effects of skin color previously reported (see Figs 2–6); however, our results

suggest that skin texture does have an effect, particularly on perceived health. Therefore, skin texture might interact with skin color in perceptions of light and dark skin, and such interactions could be an exciting avenue for further studies. In addition, unlike the authors of previous studies, we manipulated overall skin color instead of gradually changing each of its aspects (e.g., luminosity, hue) [33,35,46,53,60,82,91–94]. In our study, we aimed to test the differences in skin colors typically related to two different ethnic groups by using a broader manipulation of skin color. Nonetheless, future studies could examine how each aspect of skin color contributes to the way a population is perceived.

Finally, previous studies have shown that individuals are better at judging faces of their own ethnicity [35,95]. In this study our Mexican participants were asked to rate facial stimuli of a more closely related ethnic group (Me´Phaa faces), and stimuli from a more distant ethnic group (European American faces). Although 83% of our raters reported living in Mexico City and the surrounded area, and hence were likely exposed to European facial features due to immigration of people of European ethnicity and access to the internet and social media [96,97], these differences in the effects of the skin color manipulation between European American and Me´Phaa faces might be due to different degrees of familiarity with the facial features of the two ethnic groups.

In summary, we show that the Mexican population uses facial color to make certain social judgments. We found that a dark-skinned face is perceived as more dominant and more masculine. Those perceptions may be due to a process of coevolution between the human mind and the sex difference in skin color. One may therefore infer that male skin became darker through male-male competition rather than through female choice. We also found that a light-skinned face is perceived as more attractive, more trustworthy, and healthier. Those perceptions may be due to the association between skin color and social status that prevailed in Mexico during more than three centuries of colonial rule. Today, they are maintained by colorism. We are nonetheless speculating, and more data will be needed. Similar results could be obtained from other Latin American societies, which share a common colonial history and social structure.

This study may complement the social psychology and sociological literature on skin color-based discrimination. In Latin American countries, the distinction between indigenous and non-indigenous populations is a fluid continuum with complex relationships with skin color and socioeconomic status. In Mexico, cultural practices, sense of belonging, and regional language use are the characteristics that best describe this classification [98,99]; nonetheless, skin color still has effects on the way Mexicans perceive each other and, thus, make this distinction, mainly by promoting racial attitudes, and stereotypes. For example, Flores and Telles [25] found that in Mexicans with the same skin color, those with high income are twice as likely to be perceived and described as white than those with low income. Here we found that our raters were more willing to rate light-skinned faces higher in all three perceptions that are closely related to positive social interactions. Remarkably, even when our results come from a study designed to address different questions, they support the idea of a cultural preference for light skin color based on previous findings of studies on racial attitudes and social stratification. Therefore, different disciplines should continue to develop a better understanding of the complex effects that skin color has on several aspects of Mexican psychology and society.

## Supporting information

**S1 Fig. Participant's self-perceived skin color ratings.** PERLA scale distribution of the 700 participant's own skin color ratings. The color levels go from 1 to 10, being 1 the lightest and 10 the darkest. 11 (1.6%) picked skin tone level 1, 36 (5.1%) skin tone level 2, 42 (6%) skin tone

level 3, 240 (34.3%) skin tone level 4, 201 (28.8%) skin tone level 5, 102 (14.6%) skin tone level 6, 45 (6.4%) skin tone level 7, 14 (2%) skin tone level 8, 7 (1%) skin tone level 9, and 1 (0.1%) skin tone level 10.
(PDF)

**S1 Table. Perception estimates with 95% confidence intervals with all participants included.** Estimates (Est.) with 95% confidence intervals (CI) of the fixed predictors of the models for: (a) Perceived attractiveness, (b) Perceived trustworthiness, (c) Perceived health, (d) Perceived dominance, (e) Perceived aggressiveness, and (f) Perceived masculinity. *Note*: (↓) The dominance best fitted model did not include the skin color transformation predictor, therefore the estimates shown represent the main effect of the natural skin color group only.
(PDF)

**S2 Table. Pearson's correlation analysis between the scores of all perceptions rates of the natural versions only.** Pearson's correlation between the natural versions scores of attractiveness, trustworthiness, perceived health, dominance, masculinity, and aggressiveness. $*p<0.05$, $**p<0.01$, $***p<0.001$.
(PDF)

## Acknowledgments

The authors are grateful to Juan Sebastían Lucero Carrasquilla for his help and assistance during the facial photograph acquisition of the Me'Phaa sample. Finally, we would like to thank Reviewers for taking the time and effort necessary to review the manuscript. We sincerely appreciate all valuable comments and suggestions, which helped us to improve the quality of the manuscript.

## Author Contributions

**Conceptualization:** Jaaziel Martínez-Ramírez, Isaac G-Santoyo.

**Data curation:** Jaaziel Martínez-Ramírez.

**Formal analysis:** Jaaziel Martínez-Ramírez.

**Investigation:** Jaaziel Martínez-Ramírez, Isaac G-Santoyo.

**Methodology:** Jaaziel Martínez-Ramírez, Isaac G-Santoyo.

**Supervision:** Isaac G-Santoyo.

**Visualization:** Jaaziel Martínez-Ramírez.

**Writing – original draft:** Jaaziel Martínez-Ramírez, Isaac G-Santoyo.

**Writing – review & editing:** Jaaziel Martínez-Ramírez, David Puts, Javier Nieto, Isaac G-Santoyo.

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
