## [Decision Letter · Decision Letter 0]

3 Apr 2023

PONE-D-22-34514Effects of facial skin pigmentation on social judgments in a Mexican populationPLOS ONE

Dear Dr. G-Santoyo,

Thank you for submitting your manuscript to PLOS ONE. After careful consideration, we feel that it has merit but does not fully meet PLOS ONE’s publication criteria as it currently stands. Therefore, we invite you to submit a revised version of the manuscript that addresses the points raised during the review process.

The manuscript has been evaluated by three reviewers, and their comments are available below. The reviewers have raised a number of concerns that need attention. The main issues raised are:*Mixing of evolutionary concepts with social factors (e.g many of the examined variables have a social dimension, and/or derive from learned stereotyping, and societal constructs yet are framed from a biological evolutionary point of view) 

*Lack of contextualisation and separation of the underlying theories used in the model

*Participants are not evenly sampled across the selected populations, this limitation needs to be acknowledged*Clarification of statistical choices Could you please carefully revise the manuscript to address all comments raised? Please submit your revised manuscript by May 11 2023 11:59PM. If you will need more time than this to complete your revisions, please reply to this message or contact the journal office at plosone@plos.org. Please include the following items when submitting your revised manuscript:A rebuttal letter that responds to each point raised by the academic editor and reviewer(s). You should upload this letter as a separate file labeled 'Response to Reviewers'.A marked-up copy of your manuscript that highlights changes made to the original version. You should upload this as a separate file labeled 'Revised Manuscript with Track Changes'.An unmarked version of your revised paper without tracked changes. You should upload this as a separate file labeled 'Manuscript'.

We look forward to receiving your revised manuscript.

Kind regards,

Katrien Janin

Staff Editor

PLOS ONE

Journal Requirements:

Additional Editor Comments:

Dear Authors,

I have received three very interesting reviews, each from an expert with previous experience in this type of research. Additionally, I have conducted a similar (albeit very simple) study many years ago in Papua. With my knowledge in this area, it seems to me that the reviewers' remarks may improve the manuscript. One of the reviewers has recommended rejection of the article due to having a different theoretical background than the researchers. I believe that the reviewer's comments are important, but they do not disqualify the article. Instead, they can be used to a great extent when writing the introduction and discussion for the article. This will enrich the article with a different theoretical perspective, which I encourage the authors to consider. The other reviewers are much more positive about the work, although one of them has raised several statistical questions. Please provide responses to those questions."

Reviewers' comments:

Reviewer's Responses to Questions

**Comments to the Author**

1. Is the manuscript technically sound, and do the data support the conclusions?

Reviewer #1: No

Reviewer #2: Yes

Reviewer #3: Yes

2. Has the statistical analysis been performed appropriately and rigorously? 

Reviewer #1: No

Reviewer #2: Yes

Reviewer #3: Yes

3. Have the authors made all data underlying the findings in their manuscript fully available?

Reviewer #1: No

Reviewer #2: Yes

Reviewer #3: Yes

4. Is the manuscript presented in an intelligible fashion and written in standard English?

Reviewer #1: Yes

Reviewer #2: Yes

Reviewer #3: Yes

5. Review Comments to the Author

Reviewer #1: Review for PLOS One

Effects of Facial Skin Pigmentation on Social Judgement in a Mexican Population

This is an important topic. The increasing interest and research on the topic is largely about the growing acknowledgment of the importance of race and color in Mexican society, by scientists and the general population. However, this analysis though seems to be largely of an evolutionary biological view with only a nod to ongoing racism and discrimination. Indeed, I could not find any use of “racism” but rather the use of “colorism.” When colorism is acknowledged, the authors usually refer to historical colorism under colonialization, which is learned across generations. As if racism and discrimination no longer exist. There is no consideration of the role of a Western ideology of race and racism and particularly, how Mexican elites, narratives, media, etc, shaped an ideology of mestizaje, which downplayed racism. See a considerable and growing literature by Patricio Solis, Regina Martinez Casas, Alan Knight and several others that shows the history of race in Mexico and how individuals use color to evaluate others in the labor market and other arenas. Indeed, the authors cite the PERLA studies but seem to overlook its findings about discrimination and inequality. Contrary to their claim that a social classification system based on phenotype no longer exists (p.5), these studies show that it is very profound in Mexican society, like it is throughout the Americas. I found it odd for, example, the claim that indigenous women chose to mate with Spanish men simply because they wanted to give their offspring better chances. What about the role of power or even rape?

The authors focus on the evaluations according to particular personality dimensions. They should consider the role of stereotypes as in how indigenous people are considered ugly -score low on attractiveness (note the recent reactions to the actress Yalitza Aparicio) and stereotypes about not being trusted and being sickly (two other dimensions studied). These findings are consistent with the use of ethnic/racial stereotypes. Aggresiveness may also be according to the idea of indigenous as savages. The dimensions of masculinity and dominance are less clear. Racism and stereotypes about indigenous people should certainly be considered at least as a hypothesis, which may compete with hypotheses about social learning, etc. The color ratings of the European faces is also consistent with ratings as they may appear more indigenous.

As far as the analysis, I think it is mostly well done but I am surprised that the color or the ethnicity of the raters was not controlled (See Hill, Mark E. "Race of the interviewer and perception of skin color: Evidence from the multi-city study of urban inequality." American Sociological Review (2002): 99-108. The authors claim that the rater color was concentrated in a few colors and so they didn’t control that. I don’t understand why that should be important. Also, if raters are indigenous, they may evaluate in a different way. By the way, there seems to be a hesitancy to call the Me’Pha “indigenous.” For these Mexican raters, I imagine stereotypes are based on indigeneity and they may have little or no familiarity with that particular ethnic group.

I would refer to raters as “raters” or “participant raters rather than “participants.”

Reviewer #2: This is an interesting study that certainly deserves publication. Its main difficulty is that of distinguishing between gendered and ethnic significations of skin color. The two have long coexisted in Mexican society. The authors have tried to control for ethnic significations by varying skin color within each ethnic group. The results of that strategy are mixed. It seems to me that ethnic significations contaminated the raters’ perceptions of the European American faces to a greater degree than they contaminated the raters’ perceptions of the Me’Phaa faces.

There are a few errors:

- The reference to Green and Martin (1990) does not support footnote #27

- The adjective “perceived” should be used in all of the headings, and not just “perceived health.” All of the qualities are perceived.

- Since perceived aggressiveness did not differ significantly between the two groups of faces, it should not be presented in the Discussion section as a perceived quality that differs between the two groups.

In general, the wording is awkward. I would suggest the following corrections:

Lines 83- 93 – replace the four sentences with: “The main pigments—melanin, hemoglobin, and carotenoids—absorb and reflect different wavelengths of light within the visible spectrum. Collagen also contributes by scattering light [10]. The skin’s pigments vary not only between individuals and populations but also between men and women, perhaps because of sexual selection [11]. Natural selection initially favored darker skin as a means to protect against the harmful effects of UV radiation in the tropics. The pressure of selection then shifted toward lighter skin among those humans who spread into higher latitudes with lower levels of UV [12-14]”

Footnote #10 – add:

Edwards, E.A., and S.Q. Duntley. (1939). The pigments and color of living human skin. American Journal of Anatomy 65(1): 1-33. https://doi.org/10.1002/aja.1000650102

Line 102 – replace “others” with “other”

Line 110 – replace “preference honestly contributed to reproductive outcomes” with: “preference significantly contributes to reproductive success”

Lines 128-140 – replace with: “The sex difference is due to exposure of skin tissues to differing ratios of androgens to estrogens, particularly at puberty. Testosterone has a stronger effect than estrogen on melanin synthesis and vascularization of the upper dermis [26].”

Footnote #26 - add:

Edwards EA, Hamilton JB, Duntley SQ, Hubert G. Cutaneous vascular and pigmentary changes in castrate and eunuchoid men. Endocrinology 1941; 28(1): 119-128. doi:10.1210/endo-28-1-119

Manning JT, Bundred PE, Mather FM. Second to fourth digit ratio, sexual selection, and skin colour. Evolution and Human Behavior 2004; 25(1): 38-50. doi:10.1016/s1090-5138(03)00082-5

Footnote #27 - the reference to Green and Martin (1990) does not support the statement that “males undergo a more intense facultative pigmentation after sun exposure and retain it for longer periods than females whereas female skin lightens faster after a reduction in sun exposure.”

Replace the reference to Green and Martin (1990) with:

Harvey RG. Ecological factors in skin color variation among Papua New Guineans, American Journal of Physical Anthropology 1985; 66(4): 407-416. doi:10.1002/ajpa.1330660409

Footnotes #28-30 - add:

Frost P. Preference for darker faces in photographs at different phases of the menstrual cycle: Preliminary assessment of evidence for a hormonal relationship. Perceptual and Motor Skills 1994; 79(1): 507-14. doi:10.2466/pms.1994.79.1.507

Line 140 – replace “have” with “has”

Line 148 – replace “no-WEIRD” with “non-WEIRD”

Lines 151-152 – replace with: “In this study, we used an experimental design with a Mexican population to determine whether facial skin color can influence certain social perceptions: attractiveness, …”

Line 155 – replace “To accomplish this” with “To that end”

Line 164 – delete the extra comma after “perceptions”

Line 193 – replace “create” with “created”

Line 240 – replace “into” with “within”

Line 253 – replace “prevent” with “made sure”

Line 254 – insert “not” after “were”

Line 256 – replace “on” with “of”

Lines 259-261 – replace with: “… version. In other words, we compared natural European American faces with artificial light-skinned and dark-skinned versions and natural Me’Phaa faces with artificial light-skinned and dark-skinned versions. We could thus measure how the participants perceived differences in facial color independently of the face’s ethnicity.”

Line 264 – replace “in” with “on”

Line 272 – replace “men participants” with “the male participants”

Line 281 – replace “be” with “have been”

Line 285 – replace “in” with “on the”

Line 329 – replace “Attractiveness” with “Perceived attractiveness”

Line 330 – replace with: “Perceived attractiveness differed significantly between the two groups when they had their natural skin color”

Line 331 – replace “as” with “more”

Line 332 – replace “as” with “than”

Lines 332-333 – replace with: “Furthermore, there were effects when natural skin color was changed within each of the two groups. Lightening the Me’Phaa faces made them more attractive (…), and darkening the European American faces made them less attractive (…). Both groups became slightly more attractive when skin color and skin texture were homogenized (…).

Lines 343-344 – replace “men participants” with “the male participants”

Lines 344-345 – replace “women participants” with “did the female participants”

Table 2 - replace “Models estimates” with “Perception estimates”

- Replace “Attractiveness perception” with “Perceived attractiveness”

- Replace “Trustworthiness perception” with “Perceived trustworthiness”

- Replace “Dominance perception” with “Perceived dominance”

- Replace “Aggressiveness perception” with “Perceived aggressiveness”

- Replace “Masculinity perception” with “Perceived masculinity”

Line 357 – replace “Trustworthiness” with “Perceived trustworthiness”

Line 359 – replace “as” with “more”

Line 360 – replace “as” with “than”

Line 361-363 – replace with: “In addition, lightening the Me’Phaa faces increased their perceived trustworthiness (…), whereas darkening the European American faces had no effect”

Line 383 – replace “Perceived health” with “Perceived healthiness”

Line 384 – replace “As in” with “As with”

Lines 386-394 – replace with: “On average, natural European American faces were perceived as being 1.19 times healthier (…) than natural Me’Phaa faces (…). When facial skin color was changed within each ethnic group, the changes had different effects on perceptions of the two groups. Lightening the Me’Phaa faces increased their perceived health (…), whereas darkening the European American faces had no effect (…). Perceived health increased slightly for both groups when skin color and skin texture were homogenized.

Line 409 – replace “Dominance” with “Perceived dominance”

Lines 410-411 – replace with: “Unlike the previous qualities, perceived dominance was lower for European American faces that had their natural skin color”

Lines 413-416 – replace with: “in contrast to perceived attractiveness, all faces received a lower dominance rating from male participants (…) than from female participants (…). Finally, we were unable to determine any effect from lightening, darkening, or homogenizing the faces, since these predictors …”

Line 420 – replace “Aggressiveness” with “Perceived aggressiveness”

Lines 421- 427 – replace with: “Although perceived aggressiveness was lower for the European American faces, the difference was not significant (…). Lightening the Me’Phaa faces made them seem less aggressive (…), but darkening the European American faces also made them seem less aggressive (…). Finally, perceived aggressiveness did not change for either group when skin color and skin texture were homogenized (…)”

Line 442 – replace “Masculinity” with “Perceived masculinity”

Lines 443-450 – “As with perceived dominance, perceived masculinity was lower for natural European American faces (…) than for natural Me’Phaa faces (…). Lightening the Me’Phaa faces made them seem less masculine (…), but darkening the European American faces had no effect (…). In addition, perceived masculinity did not change for either group when skin color and skin texture were homogenized (…).”

Lines 469-472 (sentence) – replace with: “In comparison with natural Me’Phaa faces, natural European American faces were perceived not only as more attractive, healthier, and more trustworthy but also as less dominant and less masculine. They were furthermore perceived as less aggressive, but not significantly so.”

Lines 484-486 – replace with: “… Mexican participants rated light-skinned faces more highly on all three aspects of social valence (attractiveness, healthiness, trustworthiness). In contrast, they rated dark-skinned faces more highly on two aspects of social dominance (dominance, masculinity).

Line 493 – replace “well-remunerated” with “well-paid”

Lines 495-498 (sentence) – “Consequently, a learned association between a physical trait—in this case, light skin—and social deference might establish such a perception in a population over several generations”

Line 498 – replace “learning association” with “learned association”

Lines 505-507 (sentence) – replace with: “Also, in Mexico and other Latin American countries, lightness of skin correlates with years of schooling, hourly earnings, and other indicators of well-being.”

Line 511 – replace “dark skin color faces” with “dark-skinned faces”

Lines 512-513 - replace “dark skin color faces” with “dark-skinned faces”

Line 516 – replace the comma before “therefore” with a semi-colon

Line 518 – insert “sexually” before “dimorphic”

Lines 528-535 – Replace with: “For example, lightening the Me’Phaa faces made them seem more attractive, healthier, and more trustworthy. Conversely, darkening the European American faces made them seem less attractive. Other perceived qualities, however, did not change systematically when facial skin color was changed. Perceived dominance did not change when changes were made to facial skin color within either group. Lightening the Me’Phaa faces decreased their perceived masculinity and aggressiveness, but darkening the European American faces did not increase their perceived masculinity and aggressiveness.”

Lines 541-557 - Replace with: “Moreover, we not only lightened and darkened facial color but also homogenized it by removing such texture-related aspects as spots, freckles, or wrinkles […]. Our results suggest that skin texture does have an effect, particularly on perceived health. That effect, however, largely exists independently of the effects of skin color. Skin texture might nonetheless interact with skin color in perceptions of light and dark skin, and such interactions could be an exciting avenue for further studies. In addition, unlike the authors of previous studies, we manipulated overall skin color instead of gradually changing each of its aspects (e.g., luminosity, hue) […]. Future studies could examine how each aspect of skin color contributes to the way a population is perceived.”

Lines 558-574 - replace with: “In summary, we show that the Mexican population uses facial color to make certain social judgments. We found that a dark-skinned face is perceived as more dominant and more masculine. Those perceptions may be due to a process of coevolution between the human mind and the sex difference in skin color. One may therefore infer that male skin became darker through male-male competition rather than through female choice. We also found that a light-skinned face is perceived as more attractive, more trustworthy, and healthier. Those perceptions may be due to the association between skin color and social status that prevailed in Mexico during more than three centuries of colonial rule. Today, they are maintained by colorism. We are nonetheless speculating, and more data will be needed. Similar results could be obtained from other Latin American societies, which share a common colonial history and social structure.”

Reviewer #3: The study was interested in the association between skin colour and ascribed characteristics. Because of biological adaptations, colonialism, and ethnical stereotypes, it is justified to study the effect of human skin colouration on ascribing characteristics, especially in non-WEIRD countries. The authors combine a natural experiment (in Mexico, there are people(s) of different ethnicities with varying level of skin pigmentation) and artificial manipulation of skin colouration. Fortunately, this has been done using CIELab colour space, in a relatively correct way (any manipulation presents a risk of entering an artifact to the system; no way to overcome that, however).

When contrasted with faces of native Mexican pre-Colombian group Me´Phaa, non-manipulated male faces of European origin were rated as more attractive, healthier, and more trustworthy by heterosexual raters. Me´Phaa unmanipulated faces were perceived as more dominant, masculine, and aggressive looking.

There were also effects derived from experimental skin colour manipulation, as applied by the authors: Artificial lightening of the skin colour in Me´phaa has led to higher rating of attractiveness, trustworthiness, and perceived health in the sample of the study.

In the faces of European origin (‘European Americans’) the skin colour change from lighter to darker led to lower ascribed attractiveness. While the effect of skin colour manipulation was thus in the anticipated direction, it did not affect each of the considered perceived characteristics.

Overall, the study is in my opinion concise. Given the number of independent tests, it is quite surprising, nonetheless, I was able to keep in mind its whole scope while reading; the manuscript did not collapse into a list of tests without any internal reference and inter-relationship.

The overall positive impression is also promoted by detailed report on the fitted models, availability of the script and data. It seems like a good craft (although I must admit that I have not checked if the ‘necessary’ rituals of frequentist statistics have been gone through by the authors). While I recommend to study for acceptance (this being their final destiny), I still find some aspects of it problematic – and I think these should be considered at least by extending/re-writing some paragraphs within this otherwise good manuscript.

Major objection 1: Why Poisson?

First, I wonder, why the authors decided to use generalised linear model with the family set to "Poisson". The expectation that the dataset (data of the dependent variable) comes from Poisson distribution is not commented on in the manuscript. Picking 1-9 points is, in my opinion, not a count of rare events within interval (of whatever kind; how the parameters of such a distribution may be set?). While this objection may seem serious, it is not the case (given my expectations about the "intestines" of glm fit that are above the scope of the current text). Nonetheless, the authors should either justify their decision (maybe I am just wrong, and I'd appreciate being proven so, since this may positively affect my own subsequent work) or consider using different distribution for the dependent variable’s population, which has been sampled.

Virtually any setting of this parameter is unlikely to change the results substantially (empirically derived ‘rule of thumb’) and rejecting the manuscript on the ground of it would not be justified. My understanding of using glm is to link the linear model (with the application of the proper link function) to a data (dependent variable) that cannot be treated as derived from a normal distribution (within given population).

I do not deny that it is an over-generalisation to use gaussian (normal) distribution for a rating on discontinuous scale from 1 to 9. However, there are some methods, how to overcome this issue:

(0) Ignore it and go Gaussian, which you probably decided not to.

(1) To use ordered logit link function. Thus, the fact that you have ordinal dependent variable may be acknowledged. This method allows to consider both linear overall trend and eventual non-linear association across the levels of the Likert scale.

(2) Use different rating scale (0-100). In this case, set the family= argument to gaussian in glm function (or any of its analogues that consider both individual-varying and fixed-overall estimated coefficients) is not such a big "transgression". Human ratings should come from some symmetric unimodal distribution, I guess. I do not force you to so and ask the editor to not consider this note as a suggestion for the present study (no more data collection – for now, to keep 1-9 Likert scale is okay!). However, different scale (0-100) may help in the future studies.

(3) (Almost) Lastly, any kind of ‘non-parametric’ methods or methods outside the frequentist statistics may be also helpful for future studies.

(4) Last but not least: How about central limit theorem? Once treating a single male’s face rating in this set as an estimate (one of many) of the ratings ascribed to his face in general population, you may suppose that the dependent variable is of gaussian distribution.

I anticipate that you are better frequentist statistician than me. Please, consider thus adding a paragraph to better explain your decision about the fitted models. You, the authors, may be right, I just miss the explanation.

Using model selection, AIC, etc. is IMHO okay, however, the sole fact that your ‘best’ (most parsimonious model) does not, in the case of dominance, contains skin colour manipulation as a credible (significant) predictor, necessarily imply that you would not show the estimates to the reader. Since your aim is to study the association between skin colour and ascribed characteristics (and despite the study is by its nature experimental) the explored associations shall be listed as a complete set.

Major objection 2: Culture vs… and biology vs… (?)

Another serious notion that shall be considered during revisions, refer to the lines 163-173 of the manuscript. I put aside that the sentences are long and hard to follow. I have the following concerns: The authors begin by stating that lighter skin that is naturally associated with a given ethnicity may be a cue to favourable characteristic (as they are stereotypically ascribed to a face, not to its colouration itself – colour is cue to ethnicity, which affect the ascribed characteristics). However, the opposite hypothesis would read: Maybe, it does not matter if the face is European/Native American (I mean, the origin of the face) and what matters is lightness (as an exact measure). This contrasting view (while I acknowledge cannot be directly addressed in this study) is not applied here. The authors rather switch from cultural to biological explanation: once the face is darker, it is perceived as more attractive, also dominant, aggressive, etc., due to androgens. Together, this seems for me to not make much sense. Consider rewriting this paragraph (even if it is just to help slow-minded readers, which I eventually may be, to understand).

Minor notes:

"There was no effect of the skin color manipulation or color exchange [for the perceived dominance] in any group.”

This is a surprising result, given the previously identified positive association between darker skin and dominance-related characteristics like masculinity. I suggest the authors to add a paragraph in the discussion in which they comment on the result. Consider the opposite result for perceived aggressiveness, which was not significantly affected by the ethnicity in non-manipulated faces, while it was affected by skin colour manipulation in both the groups in the anticipated direction (the same applies for masculinity).

There were also some typos (maybe as much as two or three) within the manuscript. Consider not adding any other, since you are currently well below usual load of typeset errors for a manuscript.

Thank your for the option to read the manuscript. Looking forward to its updated version.

Yours sincerely,

Vojtech Fiala

6. PLOS authors have the option to publish the peer review history of their article (what does this mean?). If published, this will include your full peer review and any attached files.

Reviewer #1: **Yes: **Edward Telles

Reviewer #2: **Yes: **Peter Frost

Reviewer #3: **Yes: **Vojtech Fiala

---

## [Author Response · Author response to Decision Letter 0]

18 May 2023

Response to edito and reviewers

Effects of facial skin pigmentation on social judgments in a Mexican population

Jaaziel Martínez-Ramírez, David Puts , Javier Nieto, Isaac G-Santoyo*

*Corr. Author.

Dear Editor, 

We thank the reviewers and you for their time and the generous comments on the manuscript. We have edited the manuscript to address all of their concerns. You will find each revised comment below. A new version of the R-Markdown where you can find some new plots and analyses is also available.

Reviewer 1: Edward Telles

-This is an important topic. The increasing interest and research on the topic is largely about the growing acknowledgment of the importance of race and color in Mexican society, by scientists and the general population. However, this analysis though seems to be largely of an evolutionary biological view with only a nod to ongoing racism and discrimination. Indeed, I could not find any use of “racism” but rather the use of “colorism.” When colorism is acknowledged, the authors usually refer to historical colorism under colonialization, which is learned across generations. As if racism and discrimination no longer exist. There is no consideration of the role of a Western ideology of race and racism and particularly, how Mexican elites, narratives, media, etc, shaped an ideology of mestizaje, which downplayed racism. See a considerable and growing literature by Patricio Solis, Regina Martinez Casas, Alan Knight and several others that shows the history of race in Mexico and how individuals use color to evaluate others in the labor market and other arenas. Indeed, the authors cite the PERLA studies but seem to overlook its findings about discrimination and inequality. Contrary to their claim that a social classification system based on phenotype no longer exists (p.5), these studies show that it is very profound in Mexican society, like it is throughout the Americas. I found it odd for, example, the claim that indigenous women chose to mate with Spanish men simply because they wanted to give their offspring better chances. What about the role of power or even rape?

Response: 

We appreciate this comment. Indeed, our manuscript is in line with the perspective of the reviewer. The expression of various forms of racism in modern Mexican society is evident, leading to profound discrimination against individuals with physical attributes resembling pre-Columbian populations, with skin color being the most prominent attribute. Correspondingly, our study provides experimental evidence for the first time in a Mexican population that a relationship exists between "light" skin color and subjective preferences for favorable social traits such as attractiveness and reliability. Our manuscript aims to neither conceal nor distort this social effect; instead, it contributes to the extensive field of study in this area.

Based on the framework of behavioral ecology, social evolutionary psychology, and cultural evolution, our study aimed to test two non-mutually exclusive hypotheses explaining the effects of facial skin pigmentation on social judgments in the Mexican population. In the so-called social learning hypothesis, we proposed that this psychological mechanism underlies the preferences for lighter skin color and its association with social benefits, prestige, or deference. This association originated during the colonial period through the racial/color hierarchy system known as "castas," which placed the European colonizers at the top of the social hierarchy and the indigenous population at the bottom. 

Therefore, we propose that skin color has led to the hierarchization of social benefits within the Mexican population through an extensive learning process that originated with the conquest. This process has influenced the ancestral evolutionary function of this physical attribute, from a mechanism of filtering UV radiation or synthesizing vitamin D to being an essential factor in social selection. In line with this, we do not deny that racism, colorism, and discrimination are also outcomes of this process and that nowadays, the cultural preferences for lighter skin color and other European phenotypical features may be reinforced by these sociological mechanisms. Our explanation from an ecological/evolutionary perspective does not contradict other authors from different fields of knowledge. Instead, it complements its study, not only due to the implementation of an experimental design that modifies skin color and controls for other confounding variables but also because of the theoretical argument that unifies the same observed phenomenon of racism through its observation in one of its facets related to subjective preferences determining different types of social interaction. 

Additionally, although we did not explicitly use "racism," we considered "colorism" a more appropriate word to describe the effect observed regarding skin color, since we did not delve further into the implications of racism or other types of discrimination as it falls outside the scope of this study. Nonetheless, colorism, a form of prejudice or discrimination based on skin color, is evident, whereby individuals are treated differently based on this physical attribute, with lighter skin tones often being favored or perceived as more desirable than those with darker skin tones. Colorism can manifest both within racial and ethnic groups (as observed in our study) and between them, resulting in various negative impacts, such as reinforcing harmful stereotypes and creating disparities in education, employment, and social status. It is often interconnected with broader issues of racism and discrimination, making it essential to address in pursuing social justice and equality.

Consequently, we have extensively rewritten a significant portion of the introduction and discussion sections to clarify our stance ( for example see lines 113-131 and 502-513). . Moreover, we have expanded the information regarding previous studies demonstrating different racial features within the current Mexican population. Here, we have emphasized modifying the phrase "no longer exist" to more clearly convey that these preferences continue to permeate the psyche of the contemporary Mexican population, at least based on our evaluated sample (>700).

-The authors focus on the evaluations according to particular personality dimensions. They should consider the role of stereotypes as in how indigenous people are considered ugly -score low on attractiveness (note the recent reactions to the actress Yalitza Aparicio) and stereotypes about not being trusted and being sickly (two other dimensions studied). These findings are consistent with the use of ethnic/racial stereotypes. Aggressiveness may also be according to the idea of indigenous as savages. The dimensions of masculinity and dominance are less clear. Racism and stereotypes about indigenous people should certainly be considered at least as a hypothesis, which may compete with hypotheses about social learning, etc. The color ratings of the European faces is also consistent with ratings as they may appear more indigenous.

Response: We appreciate this comment. Indeed, our results could be closely linked to racism and stereotypes; again, our intention has not been to deny these concepts but to complement them. In this line, we do not posit racism and stereotypes as a hypothesis competing with the hypothesis of social learning but rather as a possible consequence of this historical-social learning, which determines what is deemed "good" or "bad" based on skin color.

What is particularly interesting here is that the rater's participants had no information about the ethnicity of the presented stimuli. Therefore, we filtered out any information that could promote stereotypes by the evaluators beforehand, to observe the exclusive effect of skin color. In our experiment (please see the referred web page in line 268- labneuroecologiacog.com), we did not use labels or any other indication to suggest that the face belonged to a European American or a Me'Phaa man. Hence, if a rater had any stereotype bias towards Indigenous people, it would be harder for them to recognize the face as Indigenous. We also applied masks to all facial stimuli to minimize potential confounding factors.

Consequently, our results demonstrate a shift in participants' perceptions when skin color is oppositely modified to the original ethnic group. This change shows that a light-skinned face was evaluated positively on socially favorable dimensions (such as attractiveness and trustworthiness) and negatively on "unfavorable" dimensions. This effect was regardless of the stimuli' facial shape, the rater's preexisting stereotypical prejudices, the original ethnic group of the facial stimuli, or the rater's self-perception of skin color or ethnicity. We clarified this information in the discussion, also emphasizing that the approximations of racism and stereotypes evaluated here could be, in part, a consequence of this historical-social learning.

-As far as the analysis, I think it is mostly well done but I am surprised that the color or the ethnicity of the raters was not controlled (See Hill, Mark E. "Race of the interviewer and perception of skin color: Evidence from the multi-city study of urban inequality." American Sociological Review (2002): 99-108. The authors claim that the rater color was concentrated in a few colors and so they didn’t control that. I don’t understand why that should be important. Also, if raters are indigenous, they may evaluate in a different way. By the way, there seems to be a hesitancy to call the Me’Pha “indigenous.” For these Mexican raters, I imagine stereotypes are based on indigeneity and they may have little or no familiarity with that particular ethnic group.

Response: 

The data collected from self-reported ethnicities and PERLA scale responses indicated that 84% of the raters identified as Mestizos, and 78% chose levels 4-6 on the PERLA scale. The remaining 16% and 22% (respectively) were divided among four other ethnic groups and seven skin color levels. It's worth mentioning that some combinations had very low percentages, such as the self-reported ethnicity "Afro-American" at only 0.3%. This low percentage of other ethnic groups and scores on the PERLA scale suggests that our sample is quite homogeneous. Here, variables with low representativeness, such as these two mentioned, lead to heterogeneity of variance due to the unbalanced data in the sample size of each predictor included (i.e., ethnicity with five levels and PERLA scale with 11 levels). This unbalanced data and over-parametrization are significant structural violations when using linear models (Zuur, Leno, & Elphick, 2010; Faraway, 2016). However, following the reviewer's observation, we conducted the method described by Zuur, Leno, and Elphick (2010) to investigate the possible effects of these two variables. After the model validation process, we plotted Pearson residuals of each selected model against participants' self-reported ethnicities and the PERLA scale responses for self-perceived skin color (the box plots can be found in the R Markdown, code lines 263-264 for the attractiveness plots, 455-456 for the dominance plots, 557-558 for the masculinity plots, 681-682 for the aggressiveness plots, 806-807 for the trustworthiness plots, and 936-937 for the perceived health plots). Due to the plots did not show any effect, and to avoid structural errors in the design of linear models, we did not include either of these two predictors. We briefly explained our decision in the statistical analysis section (lines 326-330).

Finally, following the reviewer's observation, we already had included self-perception of ethnicity as "Indigenous" rather than particular ethnic groups, such as "Me'Phaa"( Please see line 191).

-I would refer to raters as “raters” or “participant raters rather than “participants.”

Response: We changed the heading “Participants” with “Participant raters” in all manuscript. 

References:

 Faraway, J. J. (2016). Extending the linear model with R: generalized linear, mixed effects and nonparametric regression models. CRC press.

Zuur, A. F., Ieno, E. N., & Elphick, C. S. (2010). A protocol for data exploration to avoid common statistical problems. Methods in ecology and evolution, 1(1), 3-14.

Reviewer #2: Peter Frost 

- “Its main difficulty is that of distinguishing between gendered and ethnic significations of skin color. The two have long coexisted in Mexican society. The authors have tried to control for ethnic significations by varying skin color within each ethnic group. The results of that strategy are mixed. It seems to me that ethnic significations contaminated the raters’ perceptions of the European American faces to a greater degree than they contaminated the raters’ perceptions of the Me’Phaa faces.”

Response: 

We appreciate this precise observation. We propose two main reasons why we observed different results in the effects of skin color transformation between the European American and Me'Phaa faces, both closely related to the suggestions made by the reviewer.

The first possibility is that the participants may have rated the stimuli differently due to varying degrees of familiarity with the facial features of the two ethnic groups. Here, raters participants were asked to rate facial stimuli from a closely related ethnic group (Me'Phaa faces) and stimuli from a more distant ethnic group (European American faces). In this case, the effects on subjective perceptions were more evident when changing skin color within the more familiarized groups. However, although we found a similar effect on European American faces, it was not statistically significant in several social perceptions.

The second possibility is attributed to a potential artifact in our method to modify the skin color. In this case, the manipulation process outcomes could have differed slightly between the European American and Me'Phaa faces. To deal with this, we minimized the standard error by fitting random slopes for skin color transformation for each facial photograph ID. However, it is essential to highlight that there is always a risk of such artifacts in any manipulation process. We added a new paragraph in the discussion summarizing these possible explanations and limitations (lines 549-562).

-There are a few errors:

- The reference to Green and Martin (1990) does not support footnote #27

- The adjective “perceived” should be used in all of the headings, and not just “perceived health.” All of the qualities are perceived.

- Since perceived aggressiveness did not differ significantly between the two groups of faces, it should not be presented in the Discussion section as a perceived quality that differs between the two groups.

-In general, the wording is awkward. I would suggest the following corrections.

-Lines 83- 93 – replace the four sentences with: “The main pigments—melanin, hemoglobin, and carotenoids—absorb and reflect different wavelengths of light within the visible spectrum. Collagen also contributes by scattering light [10]. The skin’s pigments vary not only between individuals and populations but also between men and women, perhaps because of sexual selection [11]. Natural selection initially favored darker skin as a means to protect against the harmful effects of UV radiation in the tropics. The pressure of selection then shifted toward lighter skin among those humans who spread into higher latitudes with lower levels of UV [12-14]”

-Footnote #10 – add: 

Edwards, E.A., and S.Q. Duntley. (1939). The pigments and color of living human skin. American Journal of Anatomy 65(1): 1-33. https://doi.org/10.1002/aja.1000650102

-Line 102 – replace “others” with “other”

-Line 110 – replace “preference honestly contributed to reproductive outcomes” with: “preference significantly contributes to reproductive success”

-Lines 128-140 – replace with: “The sex difference is due to exposure of skin tissues to differing ratios of androgens to estrogens, particularly at puberty. Testosterone has a stronger effect than estrogen on melanin synthesis and vascularization of the upper dermis [26].”

-Footnote #26 - add:

Edwards EA, Hamilton JB, Duntley SQ, Hubert G. Cutaneous vascular and pigmentary changes in castrate and eunuchoid men. Endocrinology 1941; 28(1): 119-128. doi:10.1210/endo-28-1-119

Manning JT, Bundred PE, Mather FM. Second to fourth digit ratio, sexual selection, and skin colour. Evolution and Human Behavior 2004; 25(1): 38-50. doi:10.1016/s1090-5138(03)00082-5

-Footnote #27 - the reference to Green and Martin (1990) does not support the statement that “males undergo a more intense facultative pigmentation after sun exposure and retain it for longer periods than females whereas female skin lightens faster after a reduction in sun exposure.”

Replace the reference to Green and Martin (1990) with:

Harvey RG. Ecological factors in skin color variation among Papua New Guineans, American Journal of Physical Anthropology 1985; 66(4): 407-416. doi:10.1002/ajpa.1330660409

-Footnotes #28-30 - add:

Frost P. Preference for darker faces in photographs at different phases of the menstrual cycle: Preliminary assessment of evidence for a hormonal relationship. Perceptual and Motor Skills 1994; 79(1): 507-14. doi:10.2466/pms.1994.79.1.507

Line 140 – replace “have” with “has”

-Line 148 – replace “no-WEIRD” with “non-WEIRD”

-Lines 151-152 – replace with: “In this study, we used an experimental design with a Mexican population to determine whether facial skin color can influence certain social perceptions: attractiveness, …”

-Line 155 – replace “To accomplish this” with “To that end”

-Line 164 – delete the extra comma after “perceptions”

-Line 193 – replace “create” with “created”

-Line 240 – replace “into” with “within”

-Line 253 – replace “prevent” with “made sure”

-Line 254 – insert “not” after “were”

-Line 256 – replace “on” with “of”

-Lines 259-261 – replace with: “… version. In other words, we compared natural European American faces with artificial light-skinned and dark-skinned versions and natural Me’Phaa faces with artificial light-skinned and dark-skinned versions. We could thus measure how the participants perceived differences in facial color independently of the face’s ethnicity.”

-Line 264 – replace “in” with “on”

- 272 – replace “men participants” with “the male participants”

-Line 281 – replace “be” with “have been”

-Line 285 – replace “in” with “on the”

-Line 329 – replace “Attractiveness” with “Perceived attractiveness”

-Line 330 – replace with: “Perceived attractiveness differed significantly between the two groups when they had their natural skin color”

-Line 331 – replace “as” with “more”

-Line 332 – replace “as” with “than”

-Lines 332-333 – replace with: “Furthermore, there were effects when natural skin color was changed within each of the two groups. Lightening the Me’Phaa faces made them more attractive (…), and darkening the European American faces made them less attractive (…). Both groups became slightly more attractive when skin color and skin texture were homogenized (…).

-Lines 343-344 – replace “men participants” with “the male participants”

-Lines 344-345 – replace “women participants” with “did the female participants”

-Table 2 - replace “Models estimates” with “Perception estimates”

- Replace “Attractiveness perception” with “Perceived attractiveness”

- Replace “Trustworthiness perception” with “Perceived trustworthiness”

- Replace “Dominance perception” with “Perceived dominance”

- Replace “Aggressiveness perception” with “Perceived aggressiveness”

- Replace “Masculinity perception” with “Perceived masculinity”

-Line 357 – replace “Trustworthiness” with “Perceived trustworthiness”

-Line 359 – replace “as” with “more”

-Line 360 – replace “as” with “than”

-Line 361-363 – replace with: “In addition, lightening the Me’Phaa faces increased their perceived trustworthiness (…), whereas darkening the European American faces had no effect”

-Line 383 – replace “Perceived health” with “Perceived healthiness”

-Line 384 – replace “As in” with “As with”

-Lines 386-394 – replace with: “On average, natural European American faces were perceived as being 1.19 times healthier (…) than natural Me’Phaa faces (…). When facial skin color was changed within each ethnic group, the changes had different effects on perceptions of the two groups. Lightening the Me’Phaa faces increased their perceived health (…), whereas darkening the European American faces had no effect (…). Perceived health increased slightly for both groups when skin color and skin texture were homogenized.

-Line 409 – replace “Dominance” with “Perceived dominance”

-Lines 410-411 – replace with: “Unlike the previous qualities, perceived dominance was lower for European American faces that had their natural skin color”

-Lines 413-416 – replace with: “in contrast to perceived attractiveness, all faces received a lower dominance rating from male participants (…) than from female participants (…). Finally, we were unable to determine any effect from lightening, darkening, or homogenizing the faces, since these predictors …”

-Line 420 – replace “Aggressiveness” with “Perceived aggressiveness”

-Lines 421- 427 – replace with: “Although perceived aggressiveness was lower for the European American faces, the difference was not significant (…). Lightening the Me’Phaa faces made them seem less aggressive (…), but darkening the European American faces also made them seem less aggressive (…). Finally, perceived aggressiveness did not change for either group when skin color and skin texture were homogenized (…)”

-Line 442 – replace “Masculinity” with “Perceived masculinity”

-Lines 443-450 – “As with perceived dominance, perceived masculinity was lower for natural European American faces (…) than for natural Me’Phaa faces (…). Lightening the Me’Phaa faces made them seem less masculine (…), but darkening the European American faces had no effect (…). In addition, perceived masculinity did not change for either group when skin color and skin texture were homogenized (…).”

-Lines 469-472 (sentence) – replace with: “In comparison with natural Me’Phaa faces, natural European American faces were perceived not only as more attractive, healthier, and more trustworthy but also as less dominant and less masculine. They were furthermore perceived as less aggressive, but not significantly so.”

-Lines 484-486 – replace with: “… Mexican participants rated light-skinned faces more highly on all three aspects of social valence (attractiveness, healthiness, trustworthiness). In contrast, they rated dark-skinned faces more highly on two aspects of social dominance (dominance, masculinity).

-Line 493 – replace “well-remunerated” with “well-paid”

-Lines 495-498 (sentence) – “Consequently, a learned association between a physical trait—in this case, light skin—and social deference might establish such a perception in a population over several generations”

-Line 498 – replace “learning association” with “learned association”

-Lines 505-507 (sentence) – replace with: “Also, in Mexico and other Latin American countries, lightness of skin correlates with years of schooling, hourly earnings, and other indicators of well-being.”

-Line 511 – replace “dark skin color faces” with “dark-skinned faces”

-Lines 512-513 - replace “dark skin color faces” with “dark-skinned faces”

-Line 516 – replace the comma before “therefore” with a semi-colon

-Line 518 – insert “sexually” before “dimorphic”

-Lines 528-535 – Replace with: “For example, lightening the Me’Phaa faces made them seem more attractive, healthier, and more trustworthy. Conversely, darkening the European American faces made them seem less attractive. Other perceived qualities, however, did not change systematically when facial skin color was changed. Perceived dominance did not change when changes were made to facial skin color within either group. Lightening the Me’Phaa faces decreased their perceived masculinity and aggressiveness, but darkening the European American faces did not increase their perceived masculinity and aggressiveness.”

-Lines 541-557 - Replace with: “Moreover, we not only lightened and darkened facial color but also homogenized it by removing such texture-related aspects as spots, freckles, or wrinkles […]. Our results suggest that skin texture does have an effect, particularly on perceived health. That effect, however, largely exists independently of the effects of skin color. Skin texture might nonetheless interact with skin color in perceptions of light and dark skin, and such interactions could be an exciting avenue for further studies. In addition, unlike the authors of previous studies, we manipulated overall skin color instead of gradually changing each of its aspects (e.g., luminosity, hue) […]. Future studies could examine how each aspect of skin color contributes to the way a population is perceived.”

-Lines 558-574 - replace with: “In summary, we show that the Mexican population uses facial color to make certain social judgments. We found that a dark-skinned face is perceived as more dominant and more masculine. Those perceptions may be due to a process of coevolution between the human mind and the sex difference in skin color. One may therefore infer that male skin became darker through male-male competition rather than through female choice. We also found that a light-skinned face is perceived as more attractive, more trustworthy, and healthier. Those perceptions may be due to the association between skin color and social status that prevailed in Mexico during more than three centuries of colonial rule. Today, they are maintained by colorism. We are nonetheless speculating, and more data will be needed. Similar results could be obtained from other Latin American societies, which share a common colonial history and social structure.”

Response: 

We greatly appreciate the reviewer's effort and time in providing us with precise suggestions, which were timely and significantly improved our manuscript. Therefore, we have rewritten all sentences and added all recommended references as per the reviewer's assessment. Once again, thank you very much.

Reviewer #3: Vojtech Fiala

-The overall positive impression is also promoted by detailed report on the fitted models, availability of the script and data. It seems like a good craft (although I must admit that I have not checked if the ‘necessary’ rituals of frequentist statistics have been gone through by the authors). While I recommend to study for acceptance (this being their final destiny), I still find some aspects of it problematic – and I think these should be considered at least by extending/re-writing some paragraphs within this otherwise good manuscript.

-Major objection 1: Why Poisson?

-First, I wonder, why the authors decided to use generalised linear model with the family set to "Poisson". The expectation that the dataset (data of the dependent variable) comes from Poisson distribution is not commented on in the manuscript. Picking 1-9 points is, in my opinion, not a count of rare events within interval (of whatever kind; how the parameters of such a distribution may be set?). While this objection may seem serious, it is not the case (given my expectations about the "intestines" of glm fit that are above the scope of the current text). Nonetheless, the authors should either justify their decision (maybe I am just wrong, and I'd appreciate being proven so, since this may positively affect my own subsequent work) or consider using different distribution for the dependent variable’s population, which has been sampled.

Response: 

We appreciate this observation. We decided to use GLMMs with Poisson errors due to the nature of our data, which are discrete quantitative data derived from a Likert scale. We did not average the perception data (i.e. among participant scores for each face rated) because we consider that this data treatment could affect the measures of the central tendency of some facial perceptions; this was particularly true for the attractiveness rating distribution, where we observed a positively skewed counts-type distribution (the histograms can be found in the R Markdown, code lines 234-238). Because of this and the discrete nature of our dependent variables, we decided to use Poisson GLMMs instead of Gaussians, which we explained briefly in the statistical analysis section (lines 302-307).

Major objection 2: Culture vs… and biology vs… (?)

Another serious notion that shall be considered during revisions, refer to the lines 163-173 of the manuscript. I put aside that the sentences are long and hard to follow. I have the following concerns: The authors begin by stating that lighter skin that is naturally associated with a given ethnicity may be a cue to favorable characteristic (as they are stereotypically ascribed to a face, not to its colouration itself – colour is cue to ethnicity, which affect the ascribed characteristics). However, the opposite hypothesis would read: Maybe, it does not matter if the face is European/Native American (I mean, the origin of the face) and what matters is lightness (as an exact measure). This contrasting view (while I acknowledge cannot be directly addressed in this study) is not applied here. The authors rather switch from cultural to biological explanation: once the face is darker, it is perceived as more attractive, also dominant, aggressive, etc., due to androgens. Together, this seems for me to not make much sense. Consider rewriting this paragraph (even if it is just to help slow-minded readers, which I eventually may be, to understand). 

Response: 

Thank you for this observation. In this study, we tested three hypotheses, each of them having its null and alternative prediction. These three hypotheses could be mutually non-exclusive and, therefore, complement each other. In the first one we propose that social learning may be the psychological mechanism underlying the effects of facial skin pigmentation on the social perceptions. Here we sought to test whether light facial color ( and the opposite for dark facial color) promotes greater rate values in favorable social attributes, such as reliability, attractiveness or perceived health. This effect would be independent of the of the origin of the face, but only by the skin coloration. In our second hypothesis, we propose that these effects may rather be the results of ancestral sexual selection of an androgen-dependent trait such as skin color. Here, darker faces and not lighter would influence greater rates values in social perceptions such as attractiveness, aggressiveness, masculinity and dominance. If this is supported, we would expect that the experimental modification of the skin pigmentation would change the previous perceptions in both the European American and Me´Phaa faces. This means that, as noticed in the review, the study seeks to determine the role of skin color controlling for other facial features like shape. Finally, our third hypothesis suggests that neither lighter nor darker skin color would affect any social perception, neither in the original faces nor in stimuli with exchanged colors

We have rewritten this paragraph to make our hypothesis clearer (lines 170-186). 

Minor notes:

-"There was no effect of the skin color manipulation or color exchange [for the perceived dominance] in any group.”

This is a surprising result, given the previously identified positive association between darker skin and dominance-related characteristics like masculinity. I suggest the authors to add a paragraph in the discussion in which they comment on the result. Consider the opposite result for perceived aggressiveness, which was not significantly affected by the ethnicity in non-manipulated faces, while it was affected by skin colour manipulation in both the groups in the anticipated direction (the same applies for masculinity).

Response: As suggested by Torrance et al. (2014) it seems that the facial shape and surface features, such as skin color, contribute differently to judgements of attractiveness and dominance (See Torrance JS, Wincenciak J, Hahn AC, DeBruine LM, Jones BC (2014) The Relative Contributions of Facial Shape and Surface Information to Perceptions of Attractiveness and Dominance. PLoS ONE 9(10): e104415. doi: 10.1371/journal.pone.0104415). They found that the perception of dominance was more strongly affected by facial shape features whereas the opposite was observed for the perception of attractiveness. Yet another possibility is that the skin color and facial shape effects on some of these perceptions interact in such a way that skin color becomes relevant only when the facial shape strongly elicits one of these perceptions. This may be the case for what we observed in the perceptions of masculinity and aggressiveness, since the skin color manipulation affected mainly the Me´Phaa faces which, in turn, were the ones rated as more dominant and masculine. That would mean that skin color cues become relevant for the perception of masculine and aggressiveness only on faces with shapes that strongly elicit these perceptions. We have rewritten a paragraph in the discussion where we summarized this idea (lines 531-548).

---

## [Decision Letter · Decision Letter 1]

13 Jul 2023

PONE-D-22-34514R1Effects of facial skin pigmentation on social judgments in a Mexican populationPLOS ONE

Dear Dr. G-Santoyo,

Thank you for submitting your manuscript to PLOS ONE. After careful consideration, we feel that it has merit but does not fully meet PLOS ONE’s publication criteria as it currently stands. Therefore, we invite you to submit a revised version of the manuscript that addresses the points raised during the review process.

We look forward to receiving your revised manuscript.

Kind regards,

Kaida Xiao

Academic Editor

PLOS ONE

Reviewers' comments:

Reviewer's Responses to Questions

**Comments to the Author**

1. If the authors have adequately addressed your comments raised in a previous round of review and you feel that this manuscript is now acceptable for publication, you may indicate that here to bypass the “Comments to the Author” section, enter your conflict of interest statement in the “Confidential to Editor” section, and submit your "Accept" recommendation.

Reviewer #1: (No Response)

Reviewer #2: (No Response)

Reviewer #3: All comments have been addressed

Reviewer #4: (No Response)

2. Is the manuscript technically sound, and do the data support the conclusions?

Reviewer #1: No

Reviewer #2: Yes

Reviewer #3: Yes

Reviewer #4: Partly

3. Has the statistical analysis been performed appropriately and rigorously? 

Reviewer #1: No

Reviewer #2: Yes

Reviewer #3: Yes

Reviewer #4: Yes

4. Have the authors made all data underlying the findings in their manuscript fully available?

Reviewer #1: (No Response)

Reviewer #2: Yes

Reviewer #3: Yes

Reviewer #4: Yes

5. Is the manuscript presented in an intelligible fashion and written in standard English?

Reviewer #1: Yes

Reviewer #2: Yes

Reviewer #3: Yes

Reviewer #4: Yes

6. Review Comments to the Author

Reviewer #1: This revision is an improvement but there are still two problems. The first is correctable and I am not sure about the second:

1. The study focuses on the evolutionary role of skin color in Mexican society. However, they hardly engage with the metaphorical “elephant in the room:” current societal skin color inequality and discrimination. While the literature review acknowledges societal inequalities based on skin color, mostly in the social psychology and sociological literature, the authors do not discuss how it fits with their own findings. Except for mentioning this literature (not until pages 5-6), there is no serious engagement. How do their finding co-exist with those of racial attitudes, stereotyping and racial inequality? The abstract, for example, does not even mention that. I expect that readers would want to know how these evolutionary findings may complement (or not) the social science findings. For example, I see complementarity in how the findings are consistent with stereotypes of indigenous people.

2. My other comment is on the methods, which have implications for the substantive interpretations. Reviewers 2 and 3 also seem to have alluded to this. The authors claim that there is no information about the ethnicity of the stimuli subjects but that in one treatment, only skin color and not facial features, was changed. But skin color is the information! Darker skin color in Mexico suggests that persons are more indigenous. Note that skin color, like indigeneity, is fluid and on a continuum, especially between mestizos and indigenous people (see for example Villarreal 2013 in the American Sociological Review and Flores et al 2023 in the American Journal of Sociology).

Note that colorism is a subset of racism. It only exists because of racism. Racism has been a dominant ideology in the Western World for the past centuries, structuring how we think of others and sort them in our stratification systems. Mexican society is not exempt of course. This is the elephant in the room that needs to be engaged.

Reviewer #2: The manuscript could be better written. I have attached a corrected version of the PDF (see pages 16 to 51). Otherwise, I have no issues with the revised manuscript.

Reviewer #3: (No Response)

Reviewer #4: The topic is quite important. The present study adds to the growing body of studies addressing the importance of skin colour on facial perceptions and provides important psychophysical data collected from Mexican populations, which hasn’t been revealed before. The revision of the present paper has been improved a lot from the perspective of sociology, anthropology, and human Evolution. Whereas there are still some concerns that need to be addressed related to the colour parameter setting, image colour manipulations, data analysis, conclusions and discussions, etc. The paper could be published once the below issues are solved.

1. It’s a bit surprising that Lightness (L*) is the only parameter considered in this study whereas the aim is to investigate the influence of skin pigmentation on social judgments.

Obviously, the reason given in line 222 (lightness has a negative correlation with melanin) is not rational. In fact, melanin pigmentation is not only negatively affects the skin’s lightness, but possibly even more strongly and positively linked to skin yellowness. And it also influences skin redness. See for example the below reference. There are also other skin pigmentations. If it’s only lightness (L*) is investigated, the title and the scope should be restricted to the effect of ‘skin lightness’ not ‘skin pigmentation‘, on facial impressions. Otherwise, it’s not fair to study the influence of skin colour on facial impressions only based on L* values.

“Fruit over sunbed: Carotenoid skin colouration is found more attractive than melanin colouration,” Q. J. Exp. Psychol., vol. 68, no. 2, pp. 284–293, Feb. 2015, doi: 10.1080/17470218.2014.944194.

Kikuchi, Kumiko, et al. "Image analysis of skin colour heterogeneity focusing on skin chromophores and the age‐related changes in facial skin." Skin Research and Technology 21.2 (2015): 175-183.

2. The creation of facial stimuli is unusual. Two averaged faces were created first and then the selected eight facial stimuli were either added or subtracted from the averaged face. Is there any reason for doing that? It is also not clear how the skin colour manipulation was done (lines 233-242). As the selected eight faces have different face shapes/ features/ colours/ colour distributions from the prototype, how the colour was added or subtracted without holding the facial shape unchanged? Whether the colour change was also applied to the facial features (eyes, mouth, brows). Any process for those facial features? Why was 70 per cent particularly selected? What software was used? It would be helpful to specify the detailed process.

3. Apart from lightness (L*), the facial stimuli seem to show a large difference in yellowness (b*). Is that true? Since authors already ‘measured its skin colour values in CIELab colour space (line 218)’, it would be necessary to show the colour parameters in CLELAB colour space (including L*, a*, and b*) for all the face stimuli from both European American and Me’phaa groups, not just L*.

4. It’s not clear about the colour information of facial stimuli. Whether the facial stimuli used in this study could represent the naturally occurring skin colour variations in the two populations, European American and Me’phaa. The average colour or the prototype is not enough as the variations within the same ethnic group are usually larger than the variations across the two ethnic groups. See the below reference. Thus it would be very helpful to show the evolutionary meaningful skin colour range and variation of the real human faces from those two groups.

“Characterising the variations in ethnic skin colours: a new calibrated database for human skin,” Ski. Res. Technol., vol. 23, no. 1, pp. 21–29, Feb. 2017, doi: 10.1111/srt.12295.

5. Line 240-242: ‘The average lightness value was L* = 66.74, SD = 1.37 in the light skin colour transformed versions and L* = 52.75, SD = 2.58 in the dark skin colour transformed versions.’ Are the light versions include both light European American and light Me’phaa faces and the dark versions include both dark European American and dark Me’phaa faces? If so, the standard deviation looks very small. Does that mean the European American and Me’phaa faces have the same lightness levels after manipulations? How does that come if both their original lightness and the prototype are different? It’s better to give the actual colour distributions of the 48 face stimuli, not only the average lightness values.

6. It looks like the same facial stimuli changed not only the skin colour but also some other facial appearance traits after manipulations. E.g. in the below, the stimuli FC16, FC16co, and FC16cb are manipulated versions from the same original face, FC10, FC10cb, FC10co are from the same face. If that’s the case, it looks like the beard completely changed after the colour changed, especially for the light version, the beard disappeared in the meantime. Those appearance traits could significantly influence sexual dimorphism and are even more sexually dimorphic traits than skin colour, and thus affect social judgements, especially judgements from the other gender.

7. The facial gloss could have a big influence on the lightness measurement results based on image methods. E.g. the stimulus FI15 below seems to have a considerable area of highlight. Within these areas, the colour could be completely different from the normal skin colour. Any considerations regarding the facial gloss issue? And how was the gloss morphed into different colour versions?

8. Images were captured under different environments and camera settings. It would be good to give information on settings, such as the white balance, CCT, ISO, aperture size, shutter speed, image formats (raw or post-processed), any colour characterisation process, etc.

9. During the experiments, participants were asked to rate the face without particular restriction or focus, e.g. based on the facial skin colour. In such a case, it’s only meaningful to compare the rating of the lighter, natural, and darker versions of the same face. It’s not reasonable to compare different faces or faces of different ethnic groups as the influence of structural facial traits is tangled together. However, those structural facial traits are not quantified or included in the analysis. in fact, previous studies show colour is less important than the structural facial traits in face perceptions. Meanwhile, various facial colour cues may be involved in face perceptions. See the reference below. E.g. the skin colour variation may matter a lot (homogeneous skin is preferred), which links to the colour and texture homogenization mentioned in the current study. Are there any considerations in the data analysis related to those confounding variables in visual stimuli? “Different colour predictions of facial preference by Caucasian and Chinese observers,” Sci. Rep., vol. 12, no. 1, p. 12194, Jul. 2022, doi: 10.1038/s41598-022-15951-8.

10. In the linear mixed effect model, the face colour was coded as categorical variable. But the same colour category has different lightness levels, e.g. the natural European American face and the natural Me’phaa face have different L*, and the lighter versions of different faces have different L*... Why not use the actual lightness (L*) value as a continuous variable in the lme analysis? Actual lightness value could be a more accurate measure for assessing the effect of skin lightness on impression judgements. The fixed effect of other colour parameters, such as a* and b*, could be included in the lme analysis as well.

11. What’s the observer variation? Any evaluation of the consistency across different raters.

12. Are there any differences in the results rated by mestizo participants and European participants? Is there any effect of the participants’ ethnicity on ratings?

13. What’s the relationship between participants’ self-perceived skin colour ratings and their actual skin colour? Has their skin colour ever been measured by instruments like spectrophotometers? There might be a mismatch between self-reported and measured values.

14. It would be interesting to know whether there are any significant differences in the structural facial traits (facial shapes and facial features) between the European American and Me’phaa faces, and what are the differences. It might be good to give some evidence based on the image database used in the present study.

15. Both the abstract line 51-53 (‘We found that….faces of …were.. more attractive….’ ) and the discussion line 476-479 (‘In comparison with….’. ) gives an unreasonable conclusion. This conclusion is not justifying the effect of skin colour on facial impression but makes judgements which are out of the scope of the current study. It’s hard to tell whether those comparisons were made based on the L* or other skin colour information or texture or the structural facial traits of the two ethnic groups. From what has been written, the scope of this study is to understand the role of skin lightness on face perceptions, not to compare the attractiveness of the two different ethnic faces. And it’s not rational to make those comparisons on the bases of very different both structural facial traits and colour traits.

16. Similar issues for those statements in the results part: European American faces were rated ‘x times’ more attractive/trustworthy… than Me’phaa faces. Current data and analysis can only show the effect of skin colour, but cannot tell whether those ‘x times’ come from colour or other factors, also cannot prove the link between ‘which ethnic face is more attractive’ and ‘colour plays a role in it’.

17. Whether the current results lead to the hypothesis? Skin colour preference/social preference also includes some mainstream aesthetic criteria, e.g. tanning skin is preferred in Western countries while whitening skin is preferred in Eastern countries, which may have nothing to do with racism or discrimination. Have these factors been considered in the current study? From the current data and analysis, it seems to be hard to elucidate the convincing mechanisms or drives behind the phenomenon. More survey and investigation is needed to know the clear answer.

18. Are there any considerations related to the adaptation to the living environment? Raters might be exposed to different environments composed of a variety of coloured faces. The long-term adaptation to the environment might influence their perceptions. It might be helpful to provide some evidence or data about the variety/proportions of faces in the participants’ living environment.

19. Line 90: it would be more precise to say ‘lower and more seasonal UVR’.

7. PLOS authors have the option to publish the peer review history of their article (what does this mean?). If published, this will include your full peer review and any attached files.

Reviewer #1: No

Reviewer #2: **Yes: **Peter Frost

Reviewer #3: **Yes: **Vojtěch Fiala

Reviewer #4: No

---

## [Author Response · Author response to Decision Letter 1]

28 Aug 2023

Reviewer 1: 

Comment:

This revision is an improvement but there are still two problems. The first is correctable and I am not sure about the second:

1. The study focuses on the evolutionary role of skin color in Mexican society. However, they hardly engage with the metaphorical “elephant in the room:” current societal skin color inequality and discrimination. While the literature review acknowledges societal inequalities based on skin color, mostly in the social psychology and sociological literature, the authors do not discuss how it fits with their own findings. Except for mentioning this literature (not until pages 5-6), there is no serious engagement. How do their finding co-exist with those of racial attitudes, stereotyping and racial inequality? The abstract, for example, does not even mention that. I expect that readers would want to know how these evolutionary findings may complement (or not) the social science findings. For example, I see complementarity in how the findings are consistent with stereotypes of indigenous people.

Response:

We appreciate the previous comments from the reviewer; they have been instrumental in enhancing the manuscript's writing. Furthermore, we consider that these two recent observations provide valuable information to enhance the emphasis we have aimed about inequality and social discrimination based on skin stereotypes, and on how our evolutionary framework and results complement (or not) the social science findings.

We strongly agree that our results are consistent with previous studies in the social sciences regarding stereotypes on indigenous people. Additionally, we do not deny the current existence of societal skin color inequality and discrimination in the studied Mexican population. In fact, we propose that these behaviors are the result of these historical social pressures and cultural transmission.

Finally, the deliberate choice for alluding these concepts up until page 4 was because the initial segment of our introduction aimed to elucidate the physiological underpinnings of skin color evolution within the human species. It's important to note that this emphasis was not indicative of a lack of recognition or engagement with the prevailing issues of contemporary societal disparities related to skin color. To be more precise with these assertions, we have expanded the information in introduction (lines 101-120) and discussion (lines 569-585). In addition, we have modified the abstract to more clearly incorporate these aspects mentioned above (Lines 60-67).

Comment:

2. My other comment is on the methods, which have implications for the substantive interpretations. Reviewers 2 and 3 also seem to have alluded to this. The authors claim that there is no information about the ethnicity of the stimuli subjects but that in one treatment, only skin color and not facial features, was changed. But skin color is the information! Darker skin color in Mexico suggests that persons are more indigenous. Note that skin color, like indigeneity, is fluid and on a continuum, especially between mestizos and indigenous people (see for example Villarreal 2013 in the American Sociological Review and Flores et al 2023 in the American Journal of Sociology). 

Response: 

Thank you for this insightful observation. Indeed, this aspect holds pivotal significance for our proposition. Precisely, the aim of this study was to ascertain whether skin color serves as a conduit of information influencing social perceptions. In our study we supported this premise experimentally by altering skin color according to the facial color patterns of these two racial groups. Consequently, we propose that skin color can be employed to associate ethnicity with its learned social stereotypes, and this interplay could potentially underpin social perceptions. These two elements, coloration and stereotype, would not be intertwined without a long and robust process of cultural learning and transmission that facilitates the stereotyping of a racial group based on a distinguishing phenotypic trait (i.e., skin color). In fact, this forms the bedrock of our proposition concerning cultural transmission and social learning in the evolution of human culture. We elucidate and expand upon this information in lines 484-492. Aligned with this notion, what holds significance here is that the color-ethnicity-perception nexus has evolved as a consequence of learned cultural pressures that have delineated what is socially favorable (or unfavorable) based on an evaluation of skin color—a physical attribute rooted in a distinct evolutionary origin (i.e., environmental pressures due to UV radiation). In contemporary times, this attribute is also harnessed for categorization, stereotyping, and modulation of social perceptions. This historical social pressure revolving around skin color could furthermore give rise to components of discrimination and racism within modern Mexican society. These intricate connections have been deeply interwoven throughout the history of Latin America and disentangling them proves challenging when attempting to elucidate their compounded effects. We clarify and expand upon this information in lines 569-585. Lastly, while acknowledging that disentangling racial stereotypes from skin color presents a formidable challenge, we believe that our study could serve as useful reference for future endeavors in social sciences that delve into this issue with the necessary depth. We do encourage this vision in the paragraph previously reported (lines 569-585). We also included a paragraph to explain that skin color and ethnicity form a continuous gradient, rather than a categorical distinction as implemented in the current study. Therefore, future research is needed to ascertain the extent to which these social perceptions are influenced by the fluid continuum of these two dimensions. Please see lines 536-548.

Reviewer #2: 

Comment:

The manuscript could be better written. I have attached a corrected version of the PDF (see pages 16 to 51). Otherwise, I have no issues with the revised manuscript.

Response: 

We greatly appreciate the reviewer's effort and time in providing us with precise suggestions, which will significantly improve our manuscript. Therefore, we have rewritten all sentences. Once again, thank you very much.

Reviewer #4: 

Comment: 

1. It’s a bit surprising that Lightness (L*) is the only parameter considered in this study whereas the aim is to investigate the influence of skin pigmentation on social judgments.

Obviously, the reason given in line 222 (lightness has a negative correlation with melanin) is not rational. In fact, melanin pigmentation is not only negatively affects the skin’s lightness, but possibly even more strongly and positively linked to skin yellowness. And it also influences skin redness. See for example the below reference. There are also other skin pigmentations. If it’s only lightness (L*) is investigated, the title and the scope should be restricted to the effect of ‘skin lightness’ not ‘skin pigmentation‘, on facial impressions. Otherwise, it’s not fair to study the influence of skin colour on facial impressions only based on L* values. 

 “Fruit over sunbed: Carotenoid skin colouration is found more attractive than melanin colouration,” Q. J. Exp. Psychol., vol. 68, no. 2, pp. 284–293, Feb. 2015, doi: 10.1080/17470218.2014.944194.

Kikuchi, Kumiko, et al. "Image analysis of skin colour heterogeneity focusing on skin chromophores and the age‐related changes in facial skin." Skin Research and Technology 21.2 (2015): 175-183.

Response: 

Thank you for this comment. Skin pigmentation arises from a complex interplay of three primary pigments: melanin, hemoglobin, and carotenoids, along with other components like collagen. In our study, we focused on the dimension of skin pigmentation linked to melanin, which spans from light to dark, given its salience and significant variation among human populations. Our method involved creating prototypes based on the average skin color values of each group and then using the differences between these prototypes as a "mask" to generate the two skin color versions. Consequently, because our manipulation was based on broad average values, the resulting stimuli exhibited variations not solely in the lightness parameter but across all dimensions; in essence, since the skin color dimension we focused on is inherently embedded within the broader context of "skin pigmentation", and given that we didn't exclusively modify lightness while controlling over other parameters, we chose to use the term "skin pigmentation" instead of "skin lightness." 

On the other hand, we need to make it clear that we calculated the CIELab values to determine if the skin color in each of the two groups (natural lighter from European American faces, and natural darker from Me´phaa faces) had observable and contrasting color properties. Here, we found that color properties effectively differed between groups. However, we chose lightness as it was the color parameter that contrasted the most in our groups and it could also influence other properties (like yellowness). We explained this in the manuscript, see lines 203-2010, and in 213-217. 

Now, the statement we made in line 222(i.e., previous version line) is well-founded. Alaluf et al. (2002) compared measurements of human skin color (CIELab parameters) with biochemical measurements of melanin content, composition, and melanosome size in skin biopsies. They discovered a robust negative correlation between lightness and total melanin content, as well as a weaker yet significant negative correlation between lightness and melanosome size. Although a positive correlation between yellowness and total melanin content was identified, it appeared to be influenced by lightness. Additionally, no other substantial correlations were found between yellowness and the other biological measurements. Given these considerations, we believe that this parameter best represents the dimension we focused on in our objectives. Nevertheless, for a better understanding of the use of this parameter in our study, we removed the association of this parameter with the biochemical relationships with melanin, since, in line with the reviewer, other color parameters could be affected by this pigment, and because determining the specific relations between the three parameters of color and the social perceptions is just out of the scope of this study (Please see lines 212-217). Having said this, we also believe that investigating how each facet of skin color contributes to the perception of this population could open up intriguing avenues for further research. This notion is elaborated upon in the discussion (Line 543-548).

Comment: 

2. The creation of facial stimuli is unusual. Two averaged faces were created first and then the selected eight facial stimuli were either added or subtracted from the averaged face. Is there any reason for doing that? It is also not clear how the skin colour manipulation was done (lines 233-242). As the selected eight faces have different face shapes/ features/ colours/ colour distributions from the prototype, how the colour was added or subtracted without holding the facial shape unchanged? Whether the colour change was also applied to the facial features (eyes, mouth, brows). Any process for those facial features? Why was 70 per cent particularly selected? What software was used? It would be helpful to specify the detailed process. 

Response: 

We appreciate the comment. The software we used was Psychomorph and we did specify this information in the section named “Skin color facial stimuli” (now lines 218-220). Psychomorph has been used in numerous studies on face perception since it allows the independent manipulation of shape and color properties in facial photographs. To conduct the manipulations, the initial step involves outlining the facial contours of all photographs. Subsequently, two prototypes are fashioned, each representing the averaged values of the extreme points corresponding to the facial feature targeted for manipulation. Finally, using these two prototypes as reference, it is possible to create different versions of the same stimuli varying the facial feature based on the differences between the two prototypes (for a deeper understanding about how the software works and its details, we strongly recommend entering the following link: https://users.aber.ac.uk/bpt/jpsychomorph/). 

Now, the sum or subtraction was made applying in each face the two color “mask” of the color prototypes. This was previously described and clarified in the section “Skin color facial stimuli” (Lines 218-239). Moreover, the setting of 70% of difference was used as a rule of thumb that the software developers recommend in order to avoid visual aberrations. This percentage not only prevents these visual aberrations to happened, but also ensures a significant change in color properties among the stimuli within the observable and feasible ranges for each ethnic group. 

Finally, just like the unintended effect that occurred in texture homogenization because of the methodology we employed, a similar effect was observed in the color transformation, which extended to the eyes, mouth, and eyebrows. Our strategy to address this side effect involved comparing the ratings of the original versions with those of the corresponding versions with transformed skin color. This helped us measure the impact of the color homogenization effect. Nevertheless, we clarified this limitation in lines 531-548

Comment: 

3. Apart from lightness (L*), the facial stimuli seem to show a large difference in yellowness (b*). Is that true? Since authors already ‘measured its skin colour values in CIELab colour space (line 218)’, it would be necessary to show the colour parameters in CLELAB colour space (including L*, a*, and b*) for all the face stimuli from both European American and Me’phaa groups, not just L*.

Response: 

We appreciate this comment. As mentioned earlier, our calculation of the CIELab values primarily served as a validation method for selecting faces and for our manipulation process. In the response to comment 1 you can find all the information regarding why we used “L” as well as the lines manuscript we modified to clarify this important observation.

Additionally, emphasizing one of the three color parameters based on its significance to the study is a customary practice in facial perception research (as seen in Kleisner et al., 2017).

Comment: 

4. It’s not clear about the colour information of facial stimuli. Whether the facial stimuli used in this study could represent the naturally occurring skin colour variations in the two populations, European American and Me’phaa. The average colour or the prototype is not enough as the variations within the same ethnic group are usually larger than the variations across the two ethnic groups. See the below reference. Thus, it would be very helpful to show the evolutionary meaningful skin colour range and variation of the real human faces from those two groups. 

 “Characterising the variations in ethnic skin colours: a new calibrated database for human skin,” Ski. Res. Technol., vol. 23, no. 1, pp. 21–29, Feb. 2017, doi: 10.1111/srt.12295.

Response: 

Thanks for this comment. Although our study standardizes skin color towards an "average" color, we also included the "Natural" group; which mantain individual variations for each ethnic group (8 natural Me'phaa faces and 8 European American faces). When comparing the natural group (with individual color variations) and its corresponding "average" group, perceptions don't change significantly, except for health. This might partially tell us that intra-groups color variation has less impact than the categorical (and experimental) change from one "average" skin color to the other one (i.e., Dark to Light and vice versa). We also included random effects for each evaluated facial photograph in our models. Here, we noticed that this random effect plays an important role in perceptions, but the fixed effect (transformed groups) consistently affects these individual variations across the groups. This demonstrates a systematized effect of contrasting "skin" color groups, despite the random effect of natural variation. Additionally, the obtained average skin colors are within the typical range found in Mexican and European American populations. We've rewritten a section to clarify our rationale (lines 201-208 and 211-215).

Regarding the acknowledgement of the evolutionarily meaningful skin color range and variation of real human faces, we did provide a summary of human skin color evolution in the first section of the introduction (lines 75-90). Here, we emphasize that studying the evolution of human skin color has many aspects that have become more complex to describe the whole color diversity in human groups (e.g., microclimates, migration patterns, angles of UV radiation, etc.). Many of these factors are related to natural selection. In our case, we documented how the perception of human skin color, a trait shaped by natural selection, may have implications in other processes like sexual and social selection, as well as cultural transmission. This has already been clarified in the introduction and discussion.

Comment: 

5. Line 240-242: ‘The average lightness value was L* = 66.74, SD = 1.37 in the light skin colour transformed versions and L* = 52.75, SD = 2.58 in the dark skin colour transformed versions.’ Are the light versions including both light European American and light Me’phaa faces and the dark versions include both dark European American and dark Me’phaa faces? If so, the standard deviation looks very small. Does that mean the European American and Me’phaa faces have the same lightness levels after manipulations? How does that come if both their original lightness and the prototype are different? It’s better to give the actual colour distributions of the 48 face stimuli, not only the average lightness values.

Response: 

Thank you for bringing up this point. Effectively, our study employs a crossover design. Hence, we obtained a total of 6 groups, which include a version darker and a lighter for each natural color group. Moreover, as we've clarified before, the process of manipulating skin color entailed applying a "color mask" derived from the two skin color prototypes. Consequently, within these versions, the transformed groups with paired colors (2 lighter and 2 darker) did not show significant variation in lightness. This outcome aligns with expectations, given that the color manipulation was conducted in a similar manner for both natural color groups. Furthermore, we observe that the lightness (L) of the transformed color groups (darker and lighter) remains consistent with their corresponding natural groups, as well as with what is commonly found and naturally distributed in both populations. We have provided clarification on this topic in lines 211-215 and 228-235 to enhance comprehension.

Comment:

6. It looks like the same facial stimuli changed not only the skin colour but also some other facial appearance traits after manipulations. E.g. in the below, the stimuli FC16, FC16co, and FC16cb are manipulated versions from the same original face, FC10, FC10cb, FC10co are from the same face. If that’s the case, it looks like the beard completely changed after the colour changed, especially for the light version, the beard disappeared in the meantime. Those appearance traits could significantly influence sexual dimorphism and are even more sexually dimorphic traits than skin colour, and thus affect social judgements, especially judgements from the other gender.

Response: 

Although the effect of facial sexually dimorphic traits, such as beard, is widely acknowledged, in our study it did not seem to be a significant factor. In the next forest plot we show the perceived attractiveness random estimates for all three skin color versions for each of the 16 faces (form right to left: natural version, light skin color version, dark skin color version). 

The forest plot shows that none of the two natural versions (FC10, FC16) got significant higher or lower ratings than the main estimate (right column). It also shows that removing the beard did not affect the rates significantly in comparison with the main estimate for each version (light skin color version, middle column; dark skin color version, left column). We have performed the same exploration for all response variables, and we found a similar result for all of them except for the perceived masculinity ratings, however, even those differences were not significant. The rest of the forest plot can be found in the R Markdown code lines 364, 594, 720, 851, and 984). 

Comment:

7. The facial gloss could have a big influence on the lightness measurement results based on image methods. E.g. the stimulus FI15 below seems to have a considerable area of highlight. Within these areas, the colour could be completely different from the normal skin colour. Any considerations regarding the facial gloss issue? And how was the gloss morphed into different colour versions?

Response: 

Thank you for the observation. The skin color prototypes were created by averaging the values of 30 faces of each group. One of the reasons why we had that sample size was to minimize the effect of any kind of individual variation such as the one mentioned; moreover, we did not use the actual lightness values to test the effect of skin color. Now regarding a possible difference in the way the raters perceived this stimuli in all of its versions, a random effect exploration showed that there are no significative differences between the scores this stimuli got and the main estimate for each version. We have included foster plots that show this in the R Markdown (code lines 364, 594, 720, 851, and 984). 

Comment:

8. Images were captured under different environments and camera settings. It would be good to give information on settings, such as the white balance, CCT, ISO, aperture size, shutter speed, image formats (raw or post-processed), any colour characterization process, etc.

Response: 

We appreciate this observation. We understand the relevance of giving as much information on settings as possible, however the information written in the lines 190 – 200 is all the setting information we took. Regarding the color characterization process, we followed the color calibration methodology reported in lines 208-217 which is a common method in the facial perception studies. 

Comment:

9. During the experiments, participants were asked to rate the face without particular restriction or focus, e.g. based on the facial skin colour. In such a case, it’s only meaningful to compare the rating of the lighter, natural, and darker versions of the same face. It’s not reasonable to compare different faces or faces of different ethnic groups as the influence of structural facial traits is tangled together. However, those structural facial traits are not quantified or included in the analysis. in fact, previous studies show colour is less important than the structural facial traits in face perceptions. Meanwhile, various facial colour cues may be involved in face perceptions. See the reference below. E.g. the skin colour variation may matter a lot (homogeneous skin is preferred), which links to the colour and texture homogenization mentioned in the current study. Are there any considerations in the data analysis related to those confounding variables in visual stimuli? “Different colour predictions of facial preference by Caucasian and Chinese observers,” Sci. Rep., vol. 12, no. 1, p. 12194, Jul. 2022, doi: 10.1038/s41598-022-15951-8.

Response: 

We appreciate this comment. A complete explanation about why we compared the ratings between the natural versions of the European American and Me’Phaa faces can be found in the answer to the comment number 15. Now, regarding the considerations to the possible confounding effect of shape and color homogenization, we took different approaches. We designed a crossing over study where the skin color version is nested within the group in order to control the effect of facial shape; and we did so by creating different skin color version of the same facial stimuli. On the other hand, we are aware that one of the side effects of our protocol was the homogenization of color and texture, and we know this may have an effect on some of the perceptions we evaluated, so we compare the ratings of the natural versions with the ratings of the concordant skin color transformed version so we could have a measurement of this effect. Our results showed that, whereas skin color and texture homogenization seemed to affect some the perceptions, especially perceived health, in general the effect was either not significant or lower that the one due to skin color change. Nonetheless, we recognize this limitation and, hence, we highlight it in the discussion (lines 531-548). 

Comment:

10. In the linear mixed effect model, the face colour was coded as categorical variable. But the same colour category has different lightness levels, e.g. the natural European American face and the natural Me’phaa face have different L*, and the lighter versions of different faces have different L*... Why not use the actual lightness (L*) value as a continuous variable in the lme analysis? Actual lightness value could be a more accurate measure for assessing the effect of skin lightness on impression judgements. The fixed effect of other colour parameters, such as a* and b*, could be included in the lme analysis as well.

Response: 

We coded the skin color as a categorical variable because it is a more reasonable way to carry out the statistical analysis of a crossover research design. In addition, using lightness values as a continuous variable would present some major inconveniences for the data that we have. Firstly, continuous variables cannot be included as random factors, so we could not be able to deal with the lack of independency of our observations (the ratings come from three versions of the same faces). Secondly, since the skin color manipulation was based on average values, the lightness values´ distribution would behave in a “categorical” fashion, that is, the values would group in three points instead of being constantly distributed in the whole range of variation. The decision about what type of variable most be used is not based on the accuracy of the measurement but in the type of variable that fits with the specific needs of the research design. 

Comment:

11. What’s the observer variation? Any evaluation of the consistency across different raters. 

Of the observer variations we assessed, factors such as sex, age, sexual preference, self-perception of ethnic group, and skin color were included. These variations and their effects are reported in the section titled "Facial Perception Rates." Given that we recognize the relevance of the individual observer variation more than variables we measured them, we also decided to include “Rater” as a random factor in our mixed models. We know that, traditionally, a measure such as the Cronbach's alpha is calculated to know the participants response reliability. However, we think that our approach is better because mixed models do not only measure this sort of variance but also take it into account when they estimate the standard errors.

Comment:

12. Are there any differences in the results rated by mestizo participants and European participants? Is there any effect of the participants’ ethnicity on ratings?

Response:

We appreciate this question. We did not find that the raters’ ethnicities had an effect on their responses; and, as a matter of fact, this question was already asked by the first reviewer during the first submission and we wrote our answer with the whole process that led us to our conclusion in the statistical analysis section, lines 313-318. 

Comment:

13. What’s the relationship between participants’ self-perceived skin colour ratings and their actual skin colour? Has their skin colour ever been measured by instruments like spectrophotometers? There might be a mismatch between self-reported and measured values.

Response: 

This is an interesting question. Based on the literature available we do believe that there may be a mismatch between self-reported, reported by someone else and objective measures of skin color in Mexico and in other Latin American countries. In these regions, individuals could perceive themselves as lighter compared to others' perceptions and even lighter than what objective measures indicate. Despite the widespread theoretical consideration of this bias, to our knowledge, no studies have directly addressed this matter within Latin American populations. This highlights a significant knowledge gap that presents a substantial opportunity for investigation. With that being said, we deem this concept to extend beyond the scope of our current study, and we recognize that a completely new study is required to comprehensively examine and test this hypothesis.

Comment:

14. It would be interesting to know whether there are any significant differences in the structural facial traits (facial shapes and facial features) between the European American and Me’phaa faces, and what are the differences. It might be good to give some evidence based on the image database used in the present study.

Response: 

This is an interesting question. A recent study found that some components of facial shape do vary across populations (Kleisner et al., 2021), and ethnicity is usually a significant predictor of facial shape variation in geometric morphometric studies. Therefore, we do expect differences in the facial shape traits between the European American and Me’phaa faces, nonetheless, similar to the previous answer, we think that this idea is beyond the reach of our study. In this study we only focused on controlling the effects of shape, however, as suggested, we are currently working on a study on facial shape, using the same image database, where we sought to test the effect of shape variation on the same perceptions, and we model the facial shape configurations that best correlate with each of the perceptions in combination with skin color 

Comment:

15. Both the abstract line 51-53 (‘We found that….faces of …were.. more attractive….’ ) and the discussion line 476-479 (‘In comparison with….’. ) gives an unreasonable conclusion. This conclusion is not justifying the effect of skin colour on facial impression but makes judgements which are out of the scope of the current study. It’s hard to tell whether those comparisons were made based on the L* or other skin colour information or texture or the structural facial traits of the two ethnic groups. From what has been written, the scope of this study is to understand the role of skin lightness on face perceptions, not to compare the attractiveness of the two different ethnic faces. And it’s not rational to make those comparisons on the bases of very different both structural facial traits and colour traits.

16. Similar issues for those statements in the results part: European American faces were rated ‘x times’ more attractive/trustworthy… than Me’phaa faces. Current data and analysis can only show the effect of skin colour, but cannot tell whether those ‘x times’ come from colour or other factors, also cannot prove the link between ‘which ethnic face is more attractive’ and ‘colour plays a role in it’.

Response: 

We agree with these two reviewer comments. Indeed, the focus of this study is to provide experimental evidence of how skin color affects social perceptions in a Mexican population, not to compare social perceptions of two ethnic groups. These groups not only differ in skin color but also in many other physical and cultural attributes, as previously mentioned. To make this objective clearer, we've made changes in the lines the reviewer mentioned and through the whole manuscript, directing the interpretation towards skin color and away from other facial or cultural attributes specific to ethnicity. Please see lines 54-57, and the results section for these modifications. 

On the other hand, we used the term "x times" because it is the accurate way to describe differences when the estimated effect comes from a Poisson GLM. In this type of GLM, the effects are expected to be exponential. Therefore, when incorporating a categorical predictor, the estimates are expressed in terms of how many times it is greater or lesser compared to the baseline, instead of a fixed and continuous difference. Although we retained these terms ("x times"), we shifted the focus to skin color (evaluated in the L dimension) and not towards an "ethnic group." Thank you for this important observation.

Comment:

17. Whether the current results lead to the hypothesis? Skin colour preference/social preference also includes some mainstream aesthetic criteria, e.g. tanning skin is preferred in Western countries while whitening skin is preferred in Eastern countries, which may have nothing to do with racism or discrimination. Have these factors been considered in the current study? From the current data and analysis, it seems to be hard to elucidate the convincing mechanisms or drives behind the phenomenon. More survey and investigation is needed to know the clear answer.

Response: 

This is an interesting question. We agree with the statement that some mainstream aesthetic criteria may play a role in the skin color preferences and that preference for tanning skin in Western countries is one example of it. However, skin color-based stratification and racism were in the very foundation of Latin American societies and still have strong effects on their social dynamic. Therefore, we think this is a more plausible factor that could explain the social preferences reported here. Nonetheless, we share the opinion that more investigation is needed to elucidate the relative contribution of mainstream aesthetic criteria on skin color preferences. 

Comment:

18. Are there any considerations related to the adaptation to the living environment? Raters might be exposed to different environments composed of a variety of coloured faces. The long-term adaptation to the environment might influence their perceptions. It might be helpful to provide some evidence or data about the variety/proportions of faces in the participants’ living environment. 

Response: 

Thank you for this observation. There are some studies that suggest that people are better at judging familiar faces, such as faces within the same ethnicity. We did propose this possibility as an explanation of the differences we observed in the effects of skin color between the two groups. We have rearranged the paragraph we had previously written to make it more visible (lines 549-557). It is important to point out that our discussion comes from an approximated measurement because there is not available data that allowed us to test this more precisely. 

Comment:

19. Line 90: it would be more precise to say ‘lower and more seasonal UVR’.

Response: 

Thank you for this observation. We have changed the term. 

Alaluf, S., Atkins, D., Barrett, K., Blount, M., Carter, N., & Heath, A. (2002). The impact of epidermal melanin on objective measurements of human skin colour. Pigment Cell Research, 15(2), 119–126. https://doi.org/10.1034/j.1600-0749.2002.1o072.x

Kleisner, K., Kočnar, T., Tureček, P., Stella, D., Akoko, R. M., Třebický, V., & Havlíček, J. (2017). African and European perception of African female attractiveness. Evolution and Human Behavior, 38(6), 744–755. https://doi.org/10.1016/j.evolhumbehav.2017.07.002

Kleisner, K., Tureček, P., Roberts, S. C., Havlíček, J., Valentova, J. V., Akoko, R. M., Leongómez, J. D., Apostol, S., Varella, M. A. C., & Saribay, S. A. (2021). How and why patterns of sexual dimorphism in human faces vary across the world. Scientific Reports, 11(1), 1–14. https://doi.org/10.1038/s41598-021-85402-3

---

## [Decision Letter · Decision Letter 2]

26 Oct 2023

Effects of facial skin pigmentation on social judgments in a Mexican population

PONE-D-22-34514R2

Dear Dr. G-Santoyo,

We’re pleased to inform you that your manuscript has been judged scientifically suitable for publication and will be formally accepted for publication once it meets all outstanding technical requirements.

Kind regards,

Kaida Xiao

Academic Editor

PLOS ONE

Additional Editor Comments (optional):

Reviewers' comments:

Reviewer's Responses to Questions

**Comments to the Author**

1. If the authors have adequately addressed your comments raised in a previous round of review and you feel that this manuscript is now acceptable for publication, you may indicate that here to bypass the “Comments to the Author” section, enter your conflict of interest statement in the “Confidential to Editor” section, and submit your "Accept" recommendation.

Reviewer #2: All comments have been addressed

Reviewer #4: All comments have been addressed

2. Is the manuscript technically sound, and do the data support the conclusions?

Reviewer #2: Yes

Reviewer #4: Yes

3. Has the statistical analysis been performed appropriately and rigorously? 

Reviewer #2: Yes

Reviewer #4: Yes

4. Have the authors made all data underlying the findings in their manuscript fully available?

Reviewer #2: Yes

Reviewer #4: Yes

5. Is the manuscript presented in an intelligible fashion and written in standard English?

Reviewer #2: Yes

Reviewer #4: Yes

6. Review Comments to the Author

Reviewer #2: On reading the revised manuscript, I noticed a few more errors in syntax and grammar. I have attached a PDF with suggested corrections. The authors may do with them as they wish.

Reviewer #4: (No Response)

7. PLOS authors have the option to publish the peer review history of their article (what does this mean?). If published, this will include your full peer review and any attached files.

Reviewer #2: **Yes: **Peter Frost

Reviewer #4: **Yes: **Y LU

---

## [Editor Report · Acceptance letter]

17 Nov 2023

PONE-D-22-34514R2 

Effects of facial skin pigmentation on social judgments in a Mexican population 

Dear Dr. G-Santoyo:

I'm pleased to inform you that your manuscript has been deemed suitable for publication in PLOS ONE. Congratulations! Your manuscript is now with our production department. 

Kind regards, 

on behalf of

Dr. Kaida Xiao 

Academic Editor

PLOS ONE